# Highly active and selective oxygen reduction to H$_2$O$_2$ on boron-doped carbon for high production rates

Yang Xia[1,9], Xunhua Zhao [2,9], Chuan Xia [1,3], Zhen-Yu Wu [1], Peng Zhu [1], Jung Yoon (Timothy) Kim [1], Xiaowan Bai[2], Guanhui Gao[4], Yongfeng Hu[5], Jun Zhong[6], Yuanyue Liu[2✉] & Haotian Wang [1,4,7,8✉]

Oxygen reduction reaction towards hydrogen peroxide (H$_2$O$_2$) provides a green alternative route for H$_2$O$_2$ production, but it lacks efficient catalysts to achieve high selectivity and activity simultaneously under industrial-relevant production rates. Here we report a boron-doped carbon (B-C) catalyst which can overcome this activity-selectivity dilemma. Compared to the state-of-the-art oxidized carbon catalyst, B-C catalyst presents enhanced activity (saving more than 210 mV overpotential) under industrial-relevant currents (up to 300 mA cm$^{-2}$) while maintaining high H$_2$O$_2$ selectivity (85–90%). Density-functional theory calculations reveal that the boron dopant site is responsible for high H$_2$O$_2$ activity and selectivity due to low thermodynamic and kinetic barriers. Employed in our porous solid electrolyte reactor, the B-C catalyst demonstrates a direct and continuous generation of pure H$_2$O$_2$ solutions with high selectivity (up to 95%) and high H$_2$O$_2$ partial currents (up to ~400 mA cm$^{-2}$), illustrating the catalyst's great potential for practical applications in the future.

[1] Department of Chemical and Biomolecular Engineering, Rice University, Houston, TX, USA. [2] Texas Materials Institute and Department of Mechanical Engineering, The University of Texas at Austin, Austin, TX, USA. [3] Smalley-Curl Institute, Rice University, Houston, TX, USA. [4] Department of Materials Science and Nanoengineering, Rice University, Houston, TX, USA. [5] Department of Chemical and Biological Engineering, University of Saskatchewan, Saskatoon, SK, Canada. [6] Institute of Functional Nano & Soft Materials (FUNSOM), Jiangsu Key Laboratory for Carbon-Based Functional Materials & Devices, Soochow University, Suzhou, China. [7] Department of Chemistry, Rice University, Houston, TX, United States. [8] Azrieli Global Scholar, Canadian Institute for Advanced Research (CIFAR), Toronto, ON, Canada. [9] These authors contributed equally: Yang Xia, Xunhua Zhao. ✉email: yuanyue.liu@austin.utexas.edu; htwang@rice.edu

Hydrogen peroxide ($H_2O_2$) is one of the most crucial and fundamental chemicals due to its wide applications in different industries, including paper and pulp manufacture, disinfection, wastewater treatment, chemical synthesis, etc[1–3]. The currently used industrial method for large-scale $H_2O_2$ production is the anthraquinone cycling process, which is an indirect chemical process based on the hydrogenation of 2-alkyl-9/10-antrhaquinone with $H_2$ followed by subsequent oxidation with $O_2$ in an organic solvent[4]. This traditional method consumes significant amounts of $H_2$ and other sorts of energy, generates a large amount of organic waste, and requires complicated separation processes to obtain high-purity $H_2O_2$ for use. Moreover, the process needs centralized large infrastructures to become economically viable, which necessities transportation and storage of highly concentrated $H_2O_2$ that is hazardous. Alternative strategies to synthesize $H_2O_2$, with less energy consumption, wastes, production cost, and safety issues, are highly desired.

Electrochemical synthesis of $H_2O_2$ from oxygen reduction reaction (ORR) via a $2e^-$ transfer process has attracted intensive interests from both academia and industry in recent years[3,5–11]. Compared to traditional anthraquinone process, its advantages include mild reaction conditions under ambient temperature and pressure, renewable electricity input without $CO_2$ emissions, high-energy conversion efficiencies, green reacting precursors such as air and water[3,5,7]. As $O_2$ molecules can also be fully reduced to $H_2O$ via a $4e^-$ transfer process[12–14], which is the preferred pathway in fuel cell applications, developing highly selective and active $2e^-$-ORR catalysts is the prerequisite of this sustainable synthetic route. Metal-based catalysts including noble metals and their alloys have been demonstrated to be selective for $H_2O_2$[15–20]. However, the cost and scarcity of noble metal elements make it difficult for future large-scale deployments. In recent years, carbon-based materials with surface functionalization, especially with the oxygen functional groups, have been recognized as promising low-cost $2e^-$-ORR catalysts candidates[21–23]. Representative examples include oxidized CNT, mildly reduced GO, and defective carbon[10,24,25]. Our recently reported oxidized carbon (O–C) catalyst also showed great $H_2O_2$ selectivity of over 90%[9]. While those oxidize carbon catalysts present negligible $H_2O_2$ onset overpotentials, they usually required significant overpotentials to deliver industrial-relevant $H_2O_2$ currents (hundreds of milliamperes per centimeter square). This sluggish kinetics of the oxidized carbon materials, especially under large currents, could stem from their high surface charge transfer resistivity, which is caused by surface oxidization[26–28]. On the other hand, pure carbon catalysts without those surface oxygen functional groups exhibit low $H_2O_2$ selectivity and activity[8,9,24]. Therefore, it becomes crucial for us to find a way to tune the carbon-based $H_2O_2$ catalysts to achieve both high activity and selectivity simultaneously, especially under industrial-relevant current densities, for practical deployments of $H_2O_2$ electrosynthesis technology in the future.

We propose to overcome this activity-selectivity dilemma by doping other nonmetal elements, instead of oxygen, into carbon catalysts for improved surface charge transfer kinetics while not sacrificing the $H_2O_2$ selectivity. Nonmetal dopants in carbon materials, including B, N, P, S, have been known to be able to serve as the active sites in catalysis, or impact the electronic properties of surrounding carbon atoms by being incorporated into the carbon lattice[21,29,30]. Previous examples have been shown in different electrocatalytic reactions including $4e^-$-ORR, $CO_2$ reduction reaction ($CO_2RR$), water splitting, etc.[30–33]. Different from O species which dramatically localize the $\pi$ electron distributions on carbon surface, dopants such as B or N could help to create new catalytic active sites while not affecting much of the surface electron transports in carbon catalysts. However, it is still unclear to us whether these nonmetal dopants can help to maintain high $H_2O_2$ selectivity while delivering significant ORR current densities under small overpotentials. Several recent studies showed some dopant effects in $2e^-$-ORR[22,34,35], but they were typically focused on onset potential regions with low ORR current densities, which are still far from practical applications' requirements.

Here we report a boron-doped carbon (B–C) as highly selective and active $2e^-$-ORR to $H_2O_2$ catalyst, especially under large current densities. Compared to the state-of-the-art oxidized carbon catalyst, our B–C catalyst presents dramatically improved activity at industrial-relevant currents while not sacrificing the $H_2O_2$ selectivity, saving over 210 mV overpotentials to deliver 300 mA cm$^{-2}$ ORR current and ~85% $H_2O_2$ Faradaic efficiency (FE). This B–C catalyst also showed excellent stability by maintaining a high activity (200 mA cm$^{-2}$) and selectivity (85–90%) for a 30 h continuous electrolysis. Other nonmetal dopants including N, P, and S showed either sluggish $H_2O_2$ activity or low selectivity compared to B–C, suggesting the central role of dopants in tuning the electronic structures of carbon-based catalysts and thus $H_2O_2$ selectivity and activity. Our DFT calculations revealed the boron sites in B–C catalyst as the origin of $H_2O_2$ activity and selectivity by considering the charge effects from both thermodynamics and kinetic aspects[36–38]. We found out that the B–C system has nearly-zero overpotential from the thermodynamic point of view, while the barrier towards $2e^-$ pathway is lower than the $4e^-$ counterpart from constant-potential molecular dynamics. These results not only agreed with our experimental observations, but also shed lights on the ORR reaction mechanisms at the atomic level. We further applied the B–C catalyst into our developed $H_2O_2$ solid electrolyte reactor for a continuous and efficient generation of pure (electrolyte-free) $H_2O_2$ solutions[9] with high FEs (up to 95%) and high $H_2O_2$ partial currents (up to 400 mA cm$^{-2}$), demonstrating the catalyst's great potential for practical applications in the future.

## Results

**Catalysts preparation and characterizations.** A series of nonmetal dopants, including boron, nitrogen, phosphorous, and sulfur, were anchored on carbon support and the resulting catalysts were screened for $2e^-$-ORR performance. Due to its high surface area, low-cost and large-scale production, carbon black (CB) is chosen as the starting material for our study. Additionally, its nanoparticulate morphology can help to enhance the $O_2$ diffusion pathways to make sure that there is a high enough concentration of $O_2$ especially under large currents, which in theory is superior to other layered supports like graphene nanosheets[39]. Following the similar procedure as reported in our previous work[9], defects were first introduced on CB (to allow for dopants coordination) in $HNO_3$ under 80 °C for 24 h. The resulting powders (O–C) were washed in DI water repeatedly until the pH gets neutral and dried. The precursors for different nonmetal elements were doped into the vacancies of the activated carbon via a chemical vapor deposition process (see Methods). To minimize oxygen impacts on $H_2O_2$ performance analysis, we chose forming gas ($H_2$/Ar) as the carrier gas to maximally remove oxygen species while other nonmetal elements were doped. As a control sample, CB powder without any dopant precursors was also annealed under the same condition (see Methods, denoted as "Pure C"). Results from X-ray photoelectron spectroscopy (XPS) high-resolution scans (Supplementary Figs. 1–4, Supplementary Table 1) showed that all the nonmetal elements were successfully doped in, with atomic ratio of 0.41, 0.86, 0.34, and 0.90 at.% for boron, nitrogen, phosphorous and sulfur dopant (the resulting materials are denoted as B–C, N–C, P–C, and S–C), respectively.

To compare their intrinsic $2e^-$-ORR performance, all the four doped carbon catalysts, as well as Pure C, were first evaluated in a standard three-electrode Rotating Ring-Disc Electrode (RRDE) setup in both alkaline (0.1M KOH) and neutral (0.1M $Na_2SO_4$) solutions. A uniform, thin catalyst layer was casted onto the glassy carbon disk electrode, with a catalyst mass loading fixed at $0.1\ mg\ cm^{-2}$. The electrolyte solution was pre-saturated with oxygen for 20 min before each run of the test. The rotation speed was fixed at 1600 rpm to allow for an efficient as-generated $H_2O_2$ diffusion to the Pt ring with minimized $H_2O_2$ decomposition. Figure 1a shows the ORR polarization curves of catalysts with different dopants in RRDE in 0.1M KOH, together with the $H_2O_2$ currents detected by the Pt ring. The corresponding $H_2O_2$ molar selectivity and FE are both plotted in Fig. 1b (see Methods). We

would like to emphasize the difference between molar selectivity and FE in RRDE test for $H_2O_2$ generation, as explained by our recent comment[40]. The detailed conversion is also illustrated in Supplementary Fig. 5 and Methods. With the incorporation of different nonmetal dopants, both activity ($H_2O_2$ partial current detected by Pt ring) and selectivity of $H_2O_2$ were significantly changed compared to Pure C. Among all the four catalysts, B–C showed the best activity, with an early onset potential (defined at $0.1\ mA\ cm^{-2}$ $H_2O_2$ partial current) of 0.773 V vs. reversible hydrogen electrode (RHE), while maintaining high molar selectivity of over 85% across a broad potential window in 0.1M KOH (Fig. 1a, b). This early onset potential is superior to or comparable to most of the catalysts reported so far[8,10,23], indicating a fast kinetics with a negligible overpotential. When

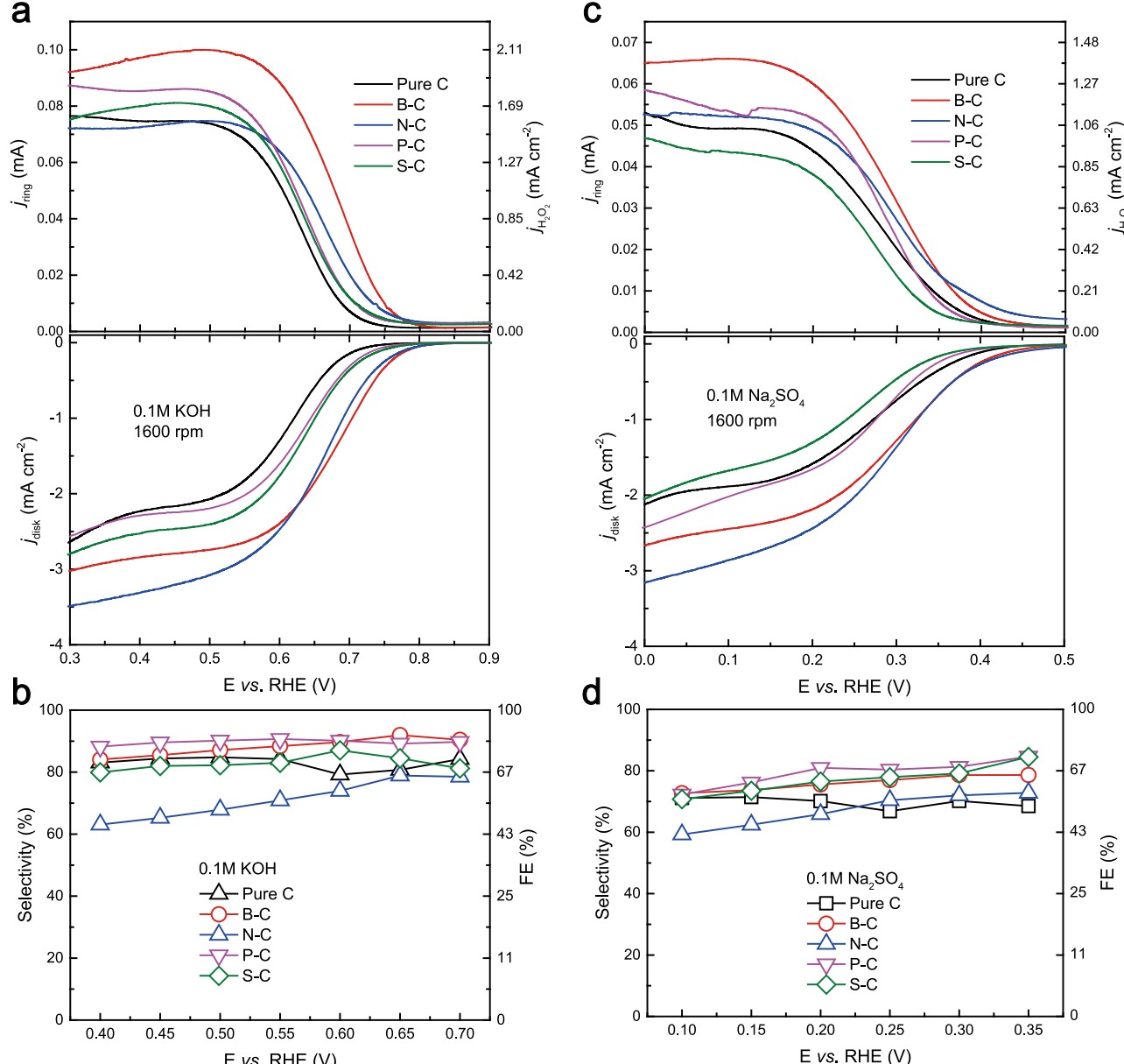

**Fig. 1 ORR performance of catalysts by RRDE.** ORR performances of catalysts by RRDE in 0.1M KOH (**a, b**) and 0.1M $Na_2SO_4$ (**c, d**). **a, c** Linear sweep voltammetry (LSV) of Pure C and B, N, P, S-doped carbon catalysts recorded at rotation rate of 1600 rpm and potential scan rate of 5 mV s$^{-1}$ (lower panel), together with the detected $H_2O_2$ currents on the Pt ring electrode (upper panel) at a fixed potential of 1.2 V vs. RHE. The corresponding $H_2O_2$ partial current densities are also plotted on the right y-axis. **b, d** $H_2O_2$ molar selectivity (left y-axis) and Faradaic efficiency (right y-axis) during the potential sweep for different catalysts in 0.1M KOH and 0.1M $Na_2SO_4$, respectively. Note that the right y-axis for Faradaic efficiency is not linearly scaled, and the detailed relation between $H_2O_2$ molar selectivity and $H_2O_2$ Faradaic efficiency is included in Supplementary Fig. 5.

the dopant was switched to other elements, the onset potential dropped to 0.771, 0.741, 0.745, and 0.71 V vs. RHE for N–C, P–C, S–C, and Pure C, respectively. Furthermore, their $H_2O_2$ partial currents also dropped compared to that of B–C. Note that even though N–C showed comparable onset potential to B–C, its $H_2O_2$ molar selectivity was only between 60 and 75%. Similar trend was also observed in neutral electrolyte (Fig. 1c, d), with B–C showing the best activity and selectivity. Such trend in RRDE suggests that the B–C sample would be the best candidate among other dopants to deliver large ORR currents under small overpotentials while maintaining high $H_2O_2$ selectivity. The RRDE performance of O–C is provided in Supplementary Fig. 6 as a reference.

The successful doping of B into the CB substrate was first confirmed by electron energy loss spectroscopy using transmission electron microscopy (TEM) (Supplementary Fig. 7). A wider range scanning of B signal was performed by the wavelength-dispersive spectroscopy (WDS) quantitative mapping analysis using the electron probe microanalyzer (EPMA) (see Methods) (Fig. 2a–c). This elemental mapping suggests a uniform distribution of B across the whole sample region. Quantitative elemental analysis shows an average of 0.60% B atomic concentration compared to bulk carbon (Supplementary Table 2), which agrees well with the XPS quantitative result. To confirm the origin of the enhanced $H_2O_2$ performance on B–C, we performed a series of characterizations on all different doping samples as well as Pure C to analyze their surface and elemental information, including morphology, surface area, surface oxygen contents as well as carbon defects. Morphological variation could directly

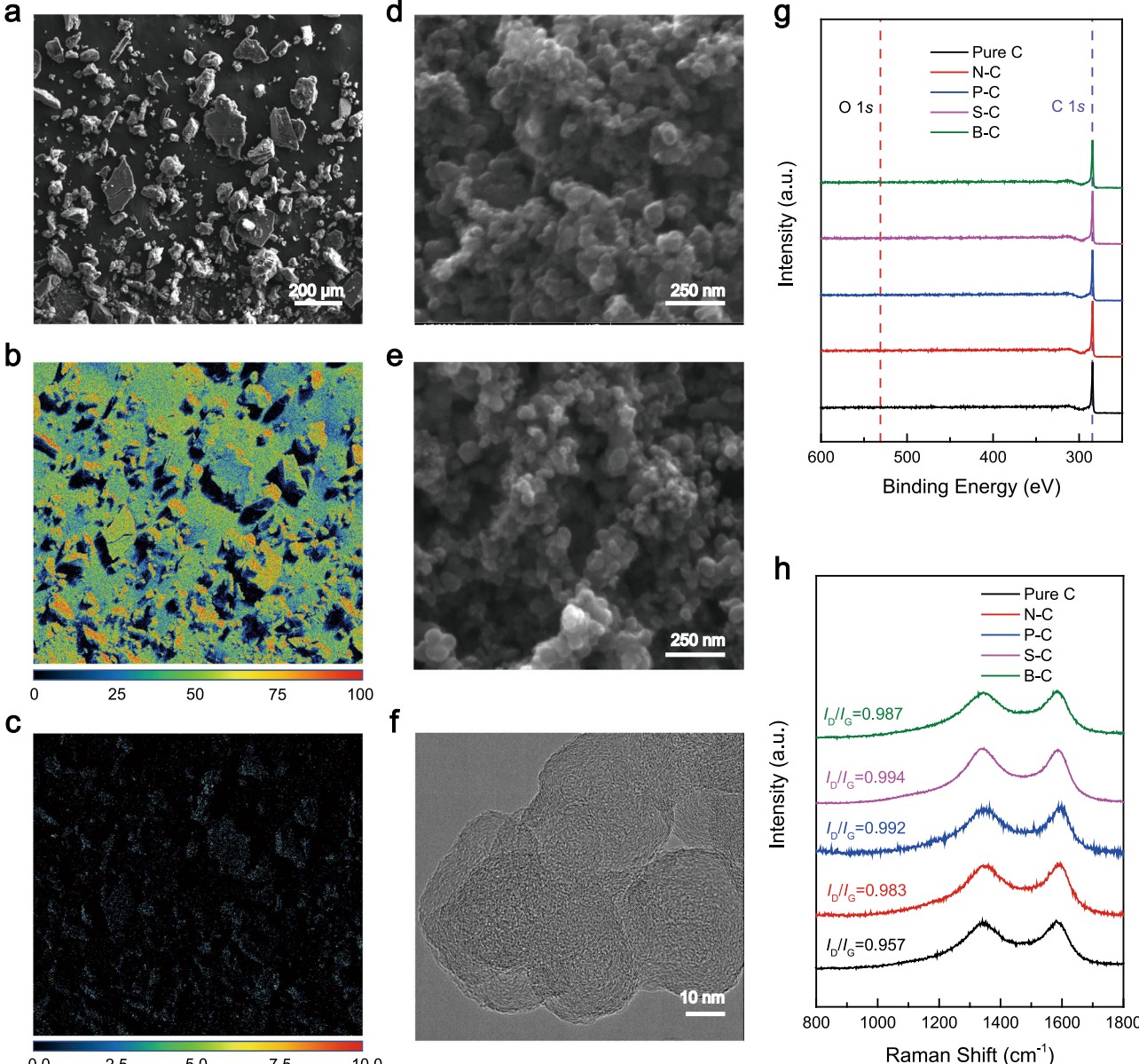

**Fig. 2 Characterizations of the catalysts. a** Back-scattered electron (BSE) image of B–C and corresponding wavelength-dispersive spectroscopy (WDS) elemental mapping for (**b**) carbon and (**c**) boron. The color bars below represent the atomic ratio (%) for each element. **d, e** Morphology of Pure C and B–C, respectively, from SEM. The scale bar represents 250 nm. **f** High-resolution TEM image of B–C, with the scale bar representing 10 nm. **g** XPS survey scan with C and O peak positions emphasis. Note that there is no obvious peak for O, indicating that our sample preparation process effectively removes the surface oxygen from the catalysts, maximally excluding the effects from surface oxygen on 2e⁻-ORR performance. **h** Raman spectroscopy for catalysts. The $I_D/I_G$ ratios are similar for all the samples, excluding the effect of defect concentration difference. Note that a.u. represents arbitrary units.

impact catalytic performances. To examine whether different dopants may induce morphological changes to the CB supports, scanning electron microscopy (SEM) and TEM were employed to analyze the sample surface morphology. All the doping samples (B–C, N–C, P–C, and S–C) showed no obvious changes from the CB support based on the SEM images in Fig. 2d, e, and Supplementary Fig. 8. High-resolution TEM images further exclude any nanometer-scale surface roughness differences between those samples (Fig. 2f, Supplementary Figs. 9–13). The intact CB morphology shown in both SEM and TEM of all the samples excludes any morphological effect on the variation of 2e$^-$-ORR performance in RRDE. The surface area of the catalyst, if subjected to any changes during different doping processes, could also play an important role in affecting catalyst's activity. To exclude or validate this impact in our B–C sample, Brunauer–Emmett–Teller (BET) analysis was employed to characterize the surface areas of all five catalytic materials as well as O–C. As a result (Supplementary Fig. 14), the BET surface area for B–C is measured to be 149.5 $m^2g^{-1}$, which is quite similar or slightly smaller to the surface areas of other dopant cases, eliminating the possibility of surface area increase that leads to the enhanced 2e$^-$-ORR activity of B–C.

A few reports in recent years demonstrated that surface oxygen functional groups can enhance the $H_2O_2$ selectivity and activity for different carbon materials, including oxidized CB, oxidized CNT, reduced graphene oxide, etc., and increased oxygen content will boost the 2e$^-$-ORR performance significantly[8–10,24]. To maximally exclude the oxygen species' impacts on our doping samples' catalytic performances, we used forming gas (5% $H_2$ in Ar) as the carrier gas during the material synthesis. Based on the XPS survey scan, there are no obvious peaks for oxygen for all the samples (Fig. 2g). A more detailed regional scan showed that the oxygen elemental ratio is quite similar for all catalysts, determined to be 0.51–1.25 at.% (Supplementary Table 1). We believe that the trace amount of oxygen came from the adsorption of water species and oxygen in air during XPS sample preparation and transportation. This proposition is confirmed by the detailed regional scan of oxygen for all the samples (Supplementary Fig. 15), in which the oxygen peaks located at 522–523 eV are contributed from adsorbed $H_2O/OH^-$, without peak observed for lattice oxygen (529–530 eV)[41]. Additional soft X-ray spectroscopy for O K-edge spectra for all the doped carbon samples (Supplementary Fig. 16) showed three peaks between 520 and 560 eV. The 1st peak around 533 eV for all of the samples is assigned to the π-bonded oxygen[42], which might come from possible trace amount of oxygen dopant which is impossible to remove when annealing the samples at the high temperature (750 °C) or formed especially on some active sites with dangling bonds once the sample is exposed to air[43–45]. The rest two peaks which dominate the O K-edge signal, one at around 535 eV and one at around 540 eV, can be assigned to residual/adsorbed water during the sample exposure to air[46]. Given the similarly low oxygen content in all the five samples, as well as the data from XAS and oxygen high-resolution analysis in XPS that we have done, we believe that it is safe to conclude that the surface oxygen is mainly coming from the adsorbed/residual water species, with a small portion of possible remaining oxygen coming from O dopants or oxidation during the sample exposure to air. The possible impacts of these trace amount of O dopants on our samples' ORR performances were successfully excluded in our later discussions. Defects in carbon materials is another factor that we need to take into consideration when we analyze the origin of $H_2O_2$ selectivity in B–C[17,25]. Raman spectroscopy was utilized to examine whether the doping process increases or decreases the concentration of defects in the CB support. As shown in Fig. 2h, similar defect

concentrations were detected for the five samples, with the $I_D/I_G$ ratios to be 0.957, 0.983, 0.992, 0.994, and 0.987 for Pure C, N–C, P–C, S–C, and B–C, respectively, indicating that the difference in 2e$^-$-ORR performance should not come from the variation of defect concentrations in different doping materials.

**DFT calculations and AIMD simulations.** From the systematic characterizations described above, we successfully excluded side factors, other than the dopant itself, that could be responsible for the significantly different 2e$^-$-ORR performances in carbon samples with different dopants. To find out the possible mechanism behind the enhanced $H_2O_2$ performance of B–C compared to the carbon support as well as other dopants, we performed density-functional theory (DFT) calculations to obtain the adsorption free energy of key intermediates in ORR and conducted ab initio molecular dynamics (AIMD) at constant potential to model the reaction kinetics. Graphene is used as the model for study because the structure of CB is comparable to graphite[30,47] (also demonstrated in the high-resolution TEM image of CB in Supplementary Fig. 17) and two-dimensional graphene would be more suitable for DFT modeling in terms of the size (more details are discussed in Supplementary Fig. 17). Specifically, three doping configurations are considered on the model of graphene in our calculations: (a) doping at single-C vacancy (SV) in graphene, (b) doping at double-C vacancy (DV), and (c) 5577 structure[48]. However, not all doping configurations are stable by themselves. For instance, the B and N tend to get out of DV hole in relaxation calculations due to their small sizes. Moreover, not all dopants create adsorption sites for *OOH, which is a key intermediate for $O_2$ reduction. Thus, we have also taken the dopant's nearest neighbor C atom into consideration as the reaction site (see Supplementary Fig. 18).

Figure 3a–d show the most favorable *OOH adsorption sites for $H_2O_2$ formation on each of the doping site, while the full table of different adsorption cases is provided in Supplementary Table 3. The free-energy diagram of the intermediate states of $O_2$ reduction for both the 2e$^-$ pathway and the 4e$^-$ pathway are shown in Fig. 3e. The potential of each intermediate state is kept at a realistic potential of 0.7 V (vs. RHE) at which the 2e$^-$ process is in thermodynamic equilibrium and the free energy is calculated as $F = E + \mu_e * n_e$ where E is the total energy of the system, $\mu_e$ represents the chemical potential of the electron at the given potential and $n_e$ is the net charge (e$^-$) of the system at $U_{RHE} = 0.7$ V (see Methods and Supplementary Fig. 19). As can be seen in Fig. 3e, it is predicted that B-doped graphene has nearly-zero overpotential towards the formation of $H_2O_2$ followed by N-doped graphene, indicating its high $H_2O_2$ formation reactivity. In contrast, P- and S- doping cases require higher overpotentials for the 2e$^-$ process. Compared among different dopants, the thermodynamics calculations at constant potential show excellent agreement with what we observed for the trend of $H_2O_2$ activity in experiments. Nevertheless, we note that quantity of over-potentials predicted for S and P are higher than what we observed in the experiment. This is probably due to the fact that only limited doping configurations were considered in our study. The free-energy diagram of the intermediate states of $O_2$ reduction for both the 2e$^-$ pathway and the 4e$^-$ pathway if each state is considered charged neutral is also provided in Supplementary Fig. 20. Note that based on this traditional method, the trend for 2e$^-$-ORR activity is still the same, but the overpotential calculated is less accurate when the actual charge is not considered for the intermediate states.

In line with results in literature[11], our results from DFT calculated thermodynamics predict the 4e$^-$ pathway being more favorable than the 2e$^-$ pathway. This is, however, in a great

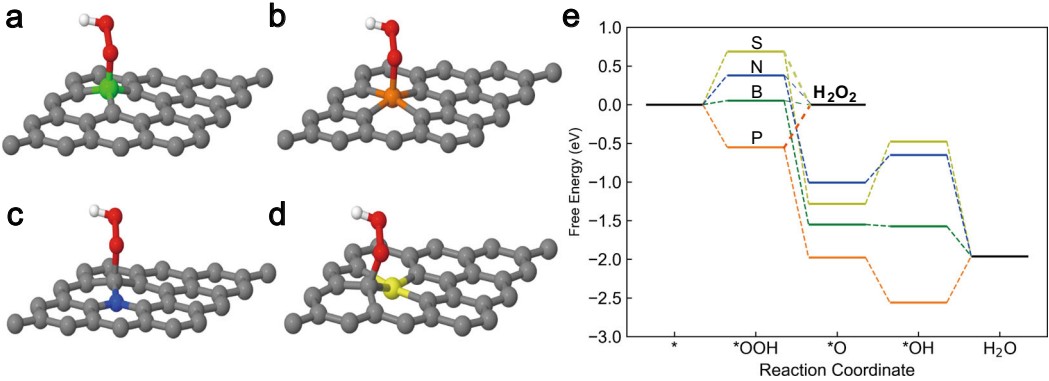

**Fig. 3 Preferred \*OOH adsorption configurations and free-energy profile by DFT studies. a–d** Preferred \*OOH adsorption configurations on B-, P-, N-, and S- doped graphene, respectively. Green, orange, blue, yellow, gray, red, and white spheres represent boron, phosphorous, nitrogen, sulphur, carbon, oxygen and hydrogen, respectively. **e** Free-energy profile of $O_2$ reduction paths where each state's charge is corresponding to the potential of $U_{RHE} = 0.7$ V (Supplementary Fig. 19).

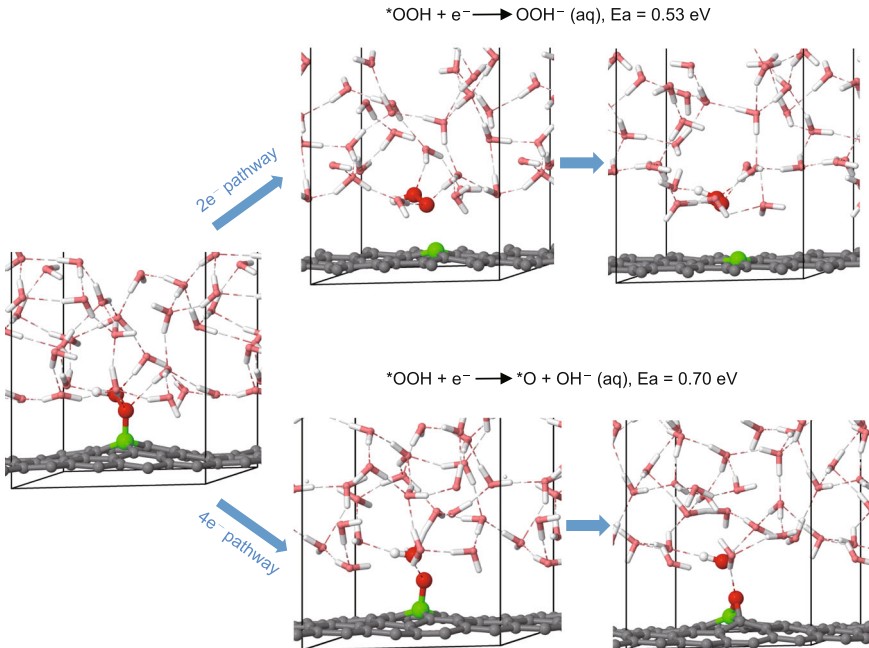

**Fig. 4 Initial snapshot (left panel), highest-free-energy snapshots (middle panels) and final snapshots (right panels) in our constant-potential MD simulations for both 2e⁻ and 4e⁻ pathways of $O_2$ reduction pathways.** The B atom is shown in green and O atoms in \*OOH are shown in red, meanwhile H atoms are shown in white and water molecules are shown in pink, respectively.

contradiction with our observation of high selectivity towards $H_2O_2$ for all the four dopants in our experiment. Knowing that the reaction barrier may give different results than thermodynamics[37], we went further to calculate the reaction barriers of the decisive steps using molecular dynamics with explicit solution (see Methods). Different from our previous study where only initial charge was adjusted[37], the charge was maintained at the constant potential all the way along the reaction path for our calculation.

As presented in Fig. 4, for the 2e⁻ pathway, OOH⁻ was formed before the formation of $H_2O_2$ and the free-energy barrier was estimated by the slow-growth approach (see Methods) to be 0.53 eV. With the presence of OOH⁻ in bulk aqueous solution, we found that the formation energy of $H_2O_2$ is only 0.07 eV. In contrast, if the reaction went towards the 4e⁻ direction, the reaction barrier of \*O formation was found to be 0.70 eV, significantly higher than that of the 2e⁻ pathway (Supplementary Fig. 21). The higher energy barrier of the formation of \*O

originated from the nature of the reaction: this reaction requires the dissociation of the strong O–O bond in which it goes through a high-energy transition state. Thus, despite the preference in terms of thermodynamics barrier calculation, the 4e⁻ pathway needs to go through a higher energy transition state, thus exhibits a lower selectivity through 4e⁻ pathway and more favorable towards $H_2O_2$ formation. These results underline that care needs to be taken when different reactions are compared with thermodynamics data only.

**Electrocatalytic 2e⁻-ORR performance of B–C.** From both RRDE screening test and DFT simulations we understand that, among different dopant samples, B–C is the best candidate catalyst to improve the overpotentials needed to deliver industrial-relevant currents while maintaining high $H_2O_2$ selectivity compared to oxidized CB catalyst (O–C). To validate our hypothesis, a standard three-electrode flow-cell reactor was used to evaluate the 2e⁻-ORR performance of our synthesized B–C catalyst in

both alkaline (1M KOH) and neutral (1M $Na_2SO_4$) solutions, together with Pure C and O–C as the control samples. The reason for using a flow-cell setup is to facilitate the $O_2$ gas diffusion to reach large-scale ORR current densities. The O–C catalyst was synthesized via the similar steps introduced in our previous work (see Methods)[9]. The flowrate of electrolyte was fixed at 54 mL h$^{-1}$ and the oxygen feed rate was set to be 20 standard cubic centimeter per minute (sccm). The saturated calomel electrode (SCE) was used as the reference electrode and a Ni foil was used as the counter electrode in the anode compartment for oxygen evolution reaction (OER). The catalysts were air-brushed on the gas diffusion layer electrode with a fixed mass loading (0.5 mg cm$^{-2}$). The concentration of generated $H_2O_2$ was determined by titration of potassium permanganate solution, and the corresponding FE was calculated based on the as-generated $H_2O_2$ concentration and the current applied (see Methods). The current–voltage (I–V) and the Faradaic efficiency–voltage (FE–V) curves of different catalysts were plotted in Fig. 5. All the I–V curves were manually iR-compensated (see Methods). In 1M KOH solution, Pure C shows the most sluggish activity compared to the other two samples, suggesting that the pure carbon surface does not contribute to significant $H_2O_2$ activity backgrounds. Although the O–C catalyst shows the most positive onset potential and excellent selectivity (over 90%), its catalytic activity was quickly surpassed by B–C under relatively large currents (>30 mA cm$^{-2}$). The advantage of B–C was further enlarged under higher current densities. As a result, B–C showed more than 100 mV less overpotential (Fig. 5a) compared to O–C in delivering 200 mA cm$^{-2}$ current, while maintaining comparably high $H_2O_2$ FE (~90%, Fig. 5b). This clearly indicates the significantly improved reaction kinetics on B–C than O–C. The $H_2O_2$ partial current in Fig. 5b illustrates the activity improvements more obviously. At a fixed applied potential of 0.70 V vs. RHE, the $H_2O_2$ partial current of B–C showed threefold improvement compared to that of O–C. Tafel plots were used to further analyze the activity of the three catalysts tested (Fig. 5c). B–C shows the best Tafel slope in alkaline condition, with the value of 78 mV/decade, which is lower than Pure C (87 mV/decade) and much lower than O–C (171 mV/decade). Similar trend was observed when switching the electrolyte to neutral condition (1M $Na_2SO_4$). B–C shows facile kinetics in large current region, with a 150 mV smaller overpotential compared to O–C at the current of 200 mA cm$^{-2}$ and 210 mV smaller at the current of 300 mA cm$^{-2}$(Fig. 5d), while maintaining comparably high FE (Fig. 5e). Higher $H_2O_2$ activities were observed in B–C compared to O–C when the total current is larger than 30 mA cm$^{-2}$. The Tafel slope of B–C is around 67 mV/decade (Fig. 5f), significantly lower than that of O–C (179 mV/decade). Compared to the state-of-the-art 2e$^-$-ORR catalysts reported in recent years, our B–C catalyst showed significant improvements especially under large current densities (Supplementary Table 4).

To better understand the intrinsic activity, the $H_2O_2$ production turnover frequency (TOF) per B site is calculated based on the electrochemical double layer capacitance. Large electrochemical surface area (ECSA)-normalized TOF on B–C shown in Supplementary Fig. 22 indicates high specific activity and catalytic efficiency of the B active center. We further compared the ECSA-normalized $H_2O_2$ activities of B–C with O–C and Pure C (Supplementary Figs. 23–25). B–C still presents similarly higher normalized $H_2O_2$ activity than Pure C and O–C, further confirming its superior intrinsic activity. Supplementary Fig. 26 shows the morphology of the three catalysts tested before and after 2e$^-$-ORR catalysis. There is no significant morphological change after the reaction compared to the unreacted samples, demonstrating the structural stability of the catalysts. Supplementary Fig. 27 shows the B high-resolution peak analysis from

XPS of B–C before and after ORR. The oxidation state of boron after the reaction does not change too much, and the boron atomic ratio kept similar (Supplementary Table 5), indicating the boron species has been kept well during ORR. In terms of morphology and defect concentrations, O–C shows similar surface morphology from SEM (Supplementary Fig. 28a) and similar $I_D/I_G$ ratio of 0.988 (Supplementary Fig. 28b) compared to B–C, excluding those effects on the performance difference shown in Fig. 5. The obvious O peak from the XPS survey scan confirms that the surface oxygen is much higher in our as-synthesized O–C sample (Supplementary Fig. 28c).

Other than its significantly improved $H_2O_2$ activities under industrial-relevant currents, stability of the B–C catalyst is another major factor to consider for its future applications in industry. We tested the stability of the B–C at different current densities, mimicking different scenarios of industrial applications (Fig. 5g and Supplementary Fig. 29). A fixed electrolyte rate of 54 mL h$^{-1}$ and a fixed $O_2$ feed rate of 20 sccm were maintained during the test. The B–C catalyst was able to retain its catalytic performance for at least 30 h without degradation in its activity and selectivity at a fixed current of 200 mA cm$^{-2}$ (Fig. 5g), suggesting its good catalytic stability even under significant production rates. B–C has also been tested in 1M $Na_2SO_4$ to show its stability under different pH conditions (Supplementary Fig. 30). A fixed electrolyte rate of 54 mL h$^{-1}$ and a fixed $O_2$ feed rate of 20 sccm were maintained during the test. The catalyst retains its catalytic activity and FE for 30 h without any degradation, indicating its excellent stability.

To further demonstrate the effect of B on the ORR performance and see whether the performance can be further improved if the B content is increased, we have slightly modified our synthesis procedures by using active carbon of higher oxidation degree (see methods) and synthesized B-doped carbon with higher B content (named as B–C-36). The B atomic ratio increased to 0.91%, while O atomic ratio slightly decreased to 0.85% (see Supplementary Table 6 for further information). The performance of the as-prepared B–C-36 sample was also tested in the standard flow-cell system. Compared to B–C, B–C-36 shows even more superior activity, with over 20 mV overpotential improvement at 200 mA cm$^{-2}$ in 1M $Na_2SO_4$ (Supplementary Fig. 31). The FE of B–C-36 is even slightly better than B–C in 1 M $Na_2SO_4$. A more direct comparison is between Pure C and B–C-36, where the oxygen level decreased from 0.99 at% to 0.85 at% (Supplementary Table 6). Obviously, with the B dopants and lower oxygen level, B–C-36 presented dramatically improved $H_2O_2$ activity (Supplementary Fig. 31). This significant improvement cannot be explained by the lower contents of oxygen dopants. Thus, we believe that it is the boron dopant that plays the central role in enhancing the 2e$^-$-ORR performance to produce $H_2O_2$, rather than the trace amount of lattice oxygen. Furthermore, we followed the same synthetic procedure of B–C-36 and synthesized another series of N–C, P–C, and S–C with higher dopant levels while the content of O nearly kept the same as previous samples, named as N–C-36, P–C-36 and S–C-36, respectively (see Supplementary Table 7 for atomic ratios). Those samples were further evaluated under high currents using the standard three-electrode flow-cell system, and the results match with our RRDE results perfectly. Specifically, in terms of activity, B–C has highest activity, followed by N–C, P–C, and S–C. Speaking of FE, B–C also shows the highest selectivity towards $H_2O_2$ production, followed by P–C, S–C, and N–C (Supplementary Fig. 32). A more convincing comparison can be made between our B–C (0.41% of B) and newly synthesized N–C-36 (1.12%), S–C-36 (1.06%), and P–C-36 (0.7%) (Supplementary Table S7), where its B doping level is lower than all of other dopants but its 2e$^-$-ORR performance is much higher than all

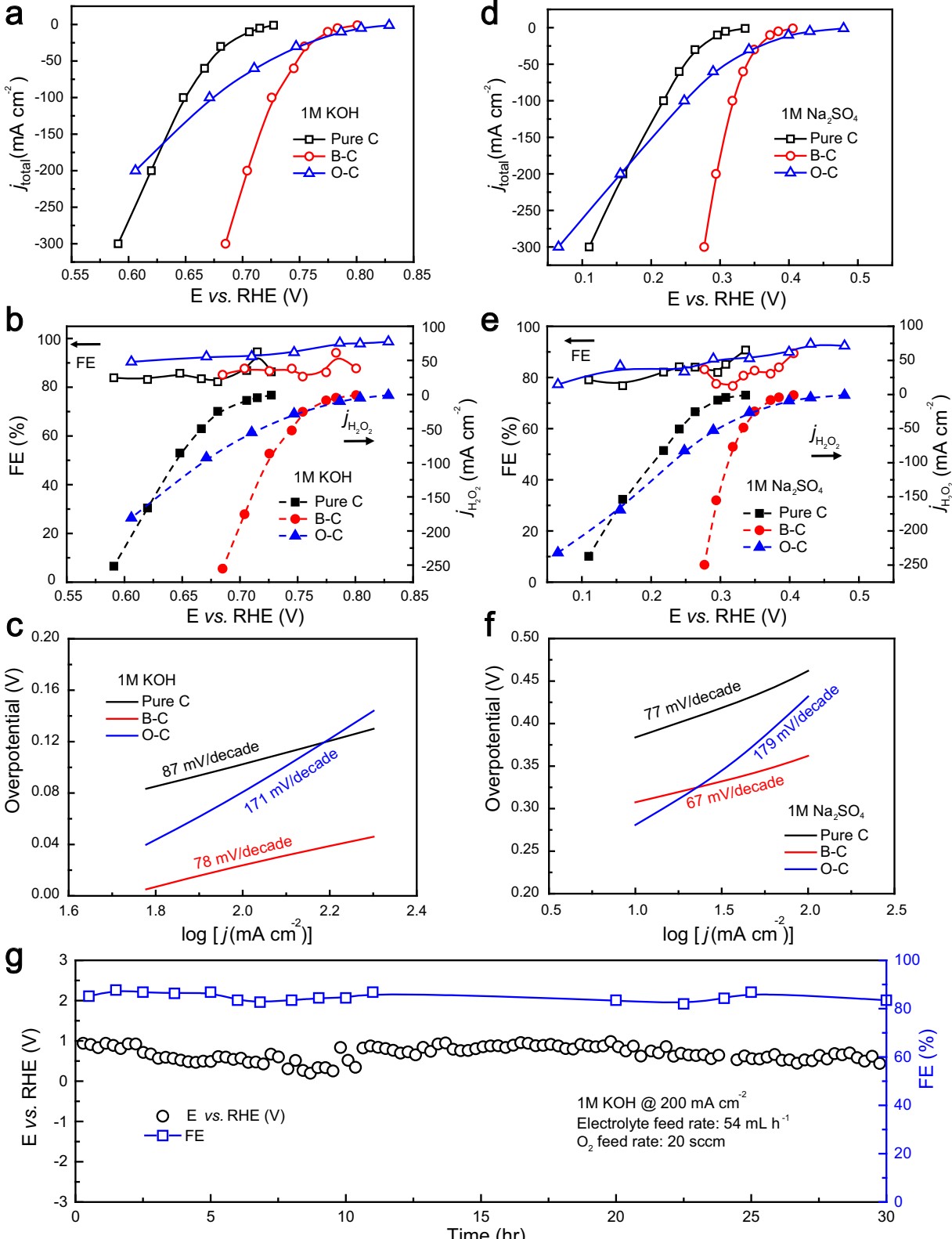

**Fig. 5 Three-electrode flow-cell performance of catalysts.** I–V curve for Pure C, B–C and O–C in (**a**) 1M KOH and (**d**) 1M Na₂SO₄. Faradaic efficiency and H₂O₂ partial currents measured are shown in (**b**) 1 M KOH and (**e**) 1 M Na₂SO₄. Note that all the I–V curves and faradaic efficiency were taken average of 2–3 independent tests for each of the samples. Corresponding Tafel plots for three catalysts in (**c**) 1 M KOH and (**f**) 1 M Na₂SO₄. The thermodynamic equilibrium potentials are taken as 0.75 V vs. RHE in 1M KOH and 0.68 V vs. RHE in 1M Na₂SO₄[19]. **g** Stability test of our B–C catalyst under 200 mA cm⁻² in 1 M KOH. The electrolyte feeding rate was fixed at 54 mL h⁻¹, and the oxygen feeding rate was fixed at 20 sccm. All the I–V curves are manually *iR*-compensated. (see Methods).

other high doping level catalysts (Supplementary Fig. 32). Thus, we can safely conclude that it is the type of dopant species, not their concentrations, that play the most important role in determining the ORR to $H_2O_2$ performances. Still, we have to mention that due to the simplicity of our synthetic procedure, it is hard to get high B content. In the future, if the synthetic procedure can be further tuned and optimized, performance of B-doped carbon can be potentially further improved with higher B content.

**Pure $H_2O_2$ generation via solid-electrolyte cell configuration.** With the facile kinetics that B–C shows in the half-cell ORR reaction, we coupled it into a whole-cell device with OER at anode side to evaluate its whole-cell performance. The problem for traditional cell configuration is its incapability to separate $H_2O_2$ product from liquid electrolyte. This makes future scaling-up undesirable because it needs expensive post-treatment processes. Continuing from the cell concept that our group demonstrated in our previous work[9], a solid-electrolyte cell (schematic shown in Fig. 6a) was used for a simple demonstration of pure $H_2O_2$ solution production for our B–C catalyst. Schematically speaking, a cation exchange membrane (CEM) and an anion exchange membrane (AEM) were used in between the solid-electrolyte middle chamber and cathode chamber and anode chamber, respectively, to avoid flooding issues of direct contact of catalyst with liquid water as well as offering selective ion transfer channels. $O_2$ can be selectively reduced to $HO_2^-$ through $2e^-$- ORR process at cathode while $H_2O$ at anode can be oxidized with an efficient OER catalyst, generating protons ($H^+$). Then, the as-generated $HO_2^-$ and $H^+$ will cross through AEM and CEM, respectively, into the solid-electrolyte layer in the middle chamber to recombine into $H_2O_2$ product which can be collected by the DI water flow. The use of solid-electrolyte is crucial, since it can offer good conductivity with small ohmic loss, as well as promoting the recombination of $HO_2^-$ and $H^+$ and introduce no ion impurity compared to the cases of traditional cell configurations. Product concentrations can be effectively tuned by controlling the DI water flow rate.

The OER catalyst that we used is the state-of-the-art $IrO_2/C$, with continuous recycling flow of 1M $H_2SO_4$. With the incorporation of our B–C catalyst into this cell configuration as the cathode, under a fixed DI water feed rate of 108 mL h$^{-1}$ and a fixed oxygen feed rate of 20 sccm, high faradaic efficiencies (>87%, up to 95%) can be reached within a broad potential window, up to current density as high as 400 mA cm$^{-2}$ (Fig. 6b). A high production rate of 7.36 mmol cm$^{-2}$ h$^{-1}$ can be achieved at 500 mA cm$^{-2}$, meanwhile about 395 mA cm$^{-2}$ $H_2O_2$ partial current can be obtained (Fig. 6c). The catalyst can undergo at least 200 h stability test for a continuous pure 1100 ppm $H_2O_2$ production without any degradation, with current fixed at 30 mA cm$^{-2}$. The FE remained stable around 90% for the entire test range, which shows excellent stability of this catalyst in the solid-electrolyte production of pure $H_2O_2$ (Fig. 6d and see Supplementary Fig. 33 for zoom-in window for the 1st 10 h). This demonstrates the excellent performance of our B–C catalyst in a real whole-cell device, making it a good candidate or a good starting catalyst to be further improved in the aspect of potential scaling-up for electrochemical $H_2O_2$ production in industrial level.

## Discussion
In summary, we have for the first time systematically studied the effects of different dopants in carbon material on its performance in $2e^-$-ORR towards $H_2O_2$. A series of characterization tools were

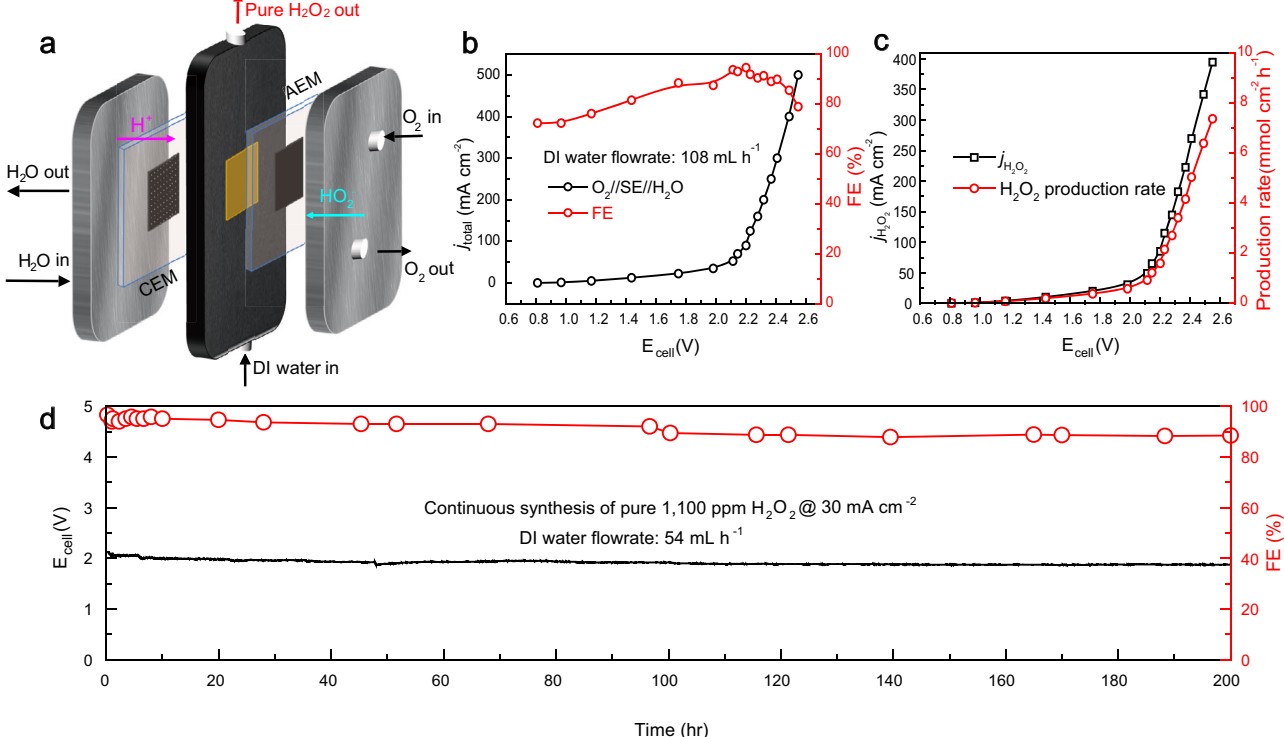

**Fig. 6 Schematic and solid-electrolyte cell performance for pure $H_2O_2$ generation. a** Schematic illustration of the solid-electrolyte cell configuration. **b** I–V curve and corresponding $H_2O_2$ faradaic efficiency. The I–V curve was manually $iR$-compensated (see Methods). **c** Corresponding $H_2O_2$ partial currents and $H_2O_2$ production rates under different applied potentials. **d** Stability test of B–C fixed at 30 mA cm$^{-2}$ of generation of ~1100 ppm pure $H_2O_2$ solution. The DI water feeding rate is fixed at 54 mL h$^{-1}$. The catalyst retains its catalytic activity and faradaic efficiency for 200 h without any degradation, indicating its excellent stability and great potential in future large-scale practical applications.

combined to study their morphological, structural, and electronic properties, excluding possible side effects on $2e^-$-ORR performance other than the dopant effect. Among all these dopants, B-doped carbon shows the highest activity and selectivity, with much enhanced kinetics compared to previously reported O–C without sacrificing the high selectivity. It presents a 210 mV overpotential improvement at 300 mA cm$^{-2}$ compared to O–C, with an excellent stability of 30-h continuous operation without degradation. DFT calculations reveals the B doped at single vacancy has nearly-zero overpotential, while molecule dynamics at constant potential indicates that the energy barrier for $2e^-$ pathway is lower than its $4e^-$ counterpart. The high-performance of B–C catalyst in our porous solid-electrolyte reactor for pure $H_2O_2$ generation demonstrated its great potential for future's practical applications. Future efforts will focus on increasing the boron doping level and its synergistic effect with other doping elements. This study solves the issues of selectivity-activity dilemma for carbon-based catalytic materials in $H_2O_2$ electrosynthesis, and provides a starting point for enhancing the $2e^-$-ORR kinetics for future energy-friendly large-scale $H_2O_2$ production.

## Methods

**Synthesis of carbon materials with different neighboring dopants.** Commercially available CB (Vulcan XC-72, Fuel Cell Store) was used as the starting carbon material. Specifically, 2 g of XC-72 was put inside a three-neck flask. Then, 460 mL 70% HNO3 solution and 140 mL DI water were added into the flask. The whole device was connected with a reflux system afterwards under well-stirred condition, and the temperature was fixed at 80 °C for 24 h. After natural cooling, the resulting slurry was centrifuged and washed with water and ethanol several times until the solution pH reaches neutral. The as-obtained precipitate was dried overnight under 80 °C in oven, and resulting sample is called O–C, as our oxidized carbon material. Different dopants, including boron, phosphorous, nitrogen and sulfur were incorporated at 750 °C under mixed H$_2$ (5%)/Ar atmosphere. Details are explained as the following:

*Pure C synthesis.* The CB powders were annealed at 750 °C under mixed H$_2$ (5%)/Ar atmosphere for 2 h (1 h temperature rising and 2 h temperature maintaining).

*Boron-doped carbon (B–C) synthesis.* The as-obtained O–C powders were mixed with boric acid (1:10 weight ratio) powders and annealed under 750 °C in Ar atmosphere for 1 h (1 h temperature rising and 2 h temperature maintaining). The resulting powder was washed by hot water for 3–4 times to remove the remaining boron oxide and dried in oven at 80 °C overnight. The remnant was further annealed under mixed H$_2$ (5%)/Ar atmosphere for 2 h under 750 °C (1 h temperature rising and 2 h temperature maintaining).

*Nitrogen-doped carbon (N–C) synthesis.* The as-obtained O–C powders were mixed with urea (1:10 weight ratio) powders and the mixture was annealed at 750 °C under mixed H$_2$ (5%)/Ar atmosphere for 2 h (1 h temperature rising and 2 h temperature maintaining).

*Phosphorous-doped carbon (P–C) synthesis.* The as-obtained O–C powders were mixed with triphenylphosphine (PPh3) (1:10 weight ratio) powders and the mixture was annealed at 750 °C under mixed H$_2$ (5%)/Ar atmosphere for 2 h (1 h temperature rising and 2 h temperature maintaining).

*Sulfur-doped carbon (S–C) Synthesis.* The as-obtained O–C powders was annealed at 750 °C, with a separate boat of sulfur powder (1:10 weight ratio) powders in the farthest upstream, under mixed H$_2$ (5%)/Ar atmosphere for 2 h (1 h of temperature rising and 2 h of temperature maintaining).

*X-C-36 series (X represents B, N, P, S) synthesis.* Commercially available CB (Vulcan XC-72, Fuel Cell Store) was used as the starting carbon material. Specifically, 0.6 g of XC-72 was put inside a three-neck flask. Then, 460 mL 70% HNO$_3$ solution and 140 mL DI water were added into the flask. The whole device was connected with a reflux system afterwards under well-stirred condition, and the temperature was fixed at 80 °C for 36 h. After natural cooling, the resulting slurry was centrifuges and washed with water and ethanol several times until the solution pH reaches neutral. The as-obtained precipitate was dried overnight under 80 °C in oven, and different dopants, including boron, phosphorous, nitrogen and sulfur were incorporated at 750 °C under mixed H$_2$ (5%)/Ar atmosphere following the same procedure as above.

**Rotation ring disk electrode (RRDE) test.** A BioLogic VMP3 workstation was used to record the electrochemical response given an applied potential. The rotation ring disk electrode (RRDE) measurements were run at RT in a typical three-electrode cell. A saturated calomel electrode (SCE, CH Instruments) was used as reference electrode. A RRDE assembly (AFE6R1PTPK, Pine Instruments) consisting of a glassy carbon rotation disk electrode (Φ = 5.0 mm) and a Pt ring (Φ = 15.0 mm) was used with a theoretical collection efficiency of 25%. Experimentally, the apparent collection efficiency (N) was determined to be 24.1% in the ferrocyanide/ferricyanide half reaction system at a rotation rate between 400 and 2025 rpm[19]. To prepare working electrode with catalyst layer, 3.3 mg of as-prepared nonmetal doping carbon catalyst was mixed with 1 mL of isopropanol and 10 μL of Nafion 117 solution (5%, Sigma-Aldrich), and sonicated for 20 min to get a homogeneous catalyst ink. 6 μL of the ink was pipetted onto glassy carbon disk (0.196 cm$^2$ surface area, and roughly 0.1 mg cm$^{-2}$ mass loading) and dried in vacuum environment. The catalyst layer showed to be uniform without uncovered edge on the glassy carbon disk electrode. All potentials measured against SCE were converted to the RHE scale in this work using E (vs. RHE) = E (vs. SCE) + 0.241 V + 0.0591 × pH, where pH values of electrolytes were determined by Orion 320 PerpHecT LogR Meter (Thermo Scientific, i.e., 13.0 for 0.1M KOH and 7.0 for 1M Na$_2$SO$_4$).

The molar selectivity of $H_2O_2$ (fraction of $O_2$ used for producing $H_2O_2$) was determined by comparing the disk current for $O_2$ reduction and $H_2O_2$ oxidation current at the Pt ring. The $H_2O_2$ molar selectivity can be calculated by equation:

$$Molar\ selectivity\ of\ H_2O_2(\%) = \frac{n_{H2O2}}{n_{H2O2} + n_{H2O}} = 200 \times \frac{i_r/N}{i_d + i_r/N} \quad (1)$$

which represents the ratio between $O_2$ consumption rate toward $H_2O_2$ and total $O_2$ consumption rate at the disk electrode.

The FE of $H_2O_2$ can be calculated by equation:

$$Faradaic\ efficiency\ of\ H_2O_2(\%) = 100 \times \frac{i_r/N}{i_d} \quad (2)$$

**Characterizations and evaluations.** All the characterizations were done at Shared Equipment Authority (SEA) at Rice University. The surface morphology was measured by SEM on a FEI Quanta 400 field emission scanning electron microscope and TEM on FEI Titan Themis S/TEM. Surface elemental analysis was performed by XPS on a PHI Quantera spectrometer, using a monochromatic Al Kα radiation (1486.6 eV) and a low-energy flood gun as the neutralizer. Note that all XPS peaks were calibrated by shifting the detected carbon C 1 s peak to 284.6 eV as the standard. Defect concentration was measured by Raman Microscope. BET surface area analysis was performed using Quantachrome Autosorb-iQMP/Kr BET Surface Analyzer. X-ray absorption spectroscopy (XAS) experiments were performed at the National Synchrotron Radiation Laboratory (NSRL, Beamlines MCD-A and MCD-B (Soochow Beamline for Energy Materials))

*Electron probe microanalysis (EPMA) analysis.* The sample powder was mounted in resin (EpoxyCure – Buehler). After cure, the sample was polished in several stages, using 600-, 800-, and 1200-grit silicon carbide paper (2 min each stage). The final polishing was done using a 1-micron gran-size of Al$_2$O$_3$ powder for 5 min. The sample was then coated with a thin film of carbon (25 nm) in order to ensure the conductivity of the electrons during the electron beam exposure. The sample and the standard specimen were coated at the same time, to ensure the same thickness of carbon film.

Electron probe microanalysis (EPMA) data acquisition consisted of a set of combined techniques: back-scattered electron imaging, semiquantitative WDS analysis, quantitative WDS point analysis, and WDS quantitative element mapping. The analysis was carried out at Rice University, Department of Earth, Environment and Planetary Science, EPMA laboratory, using a Jeol JXA 8530F Hyperprobe, equipped with a field emission assisted thermo-ionic (Schottky) emitter, and five wavelength-dispersive spectrometers. The kα peaks of boron and carbon were analyzed using specific light-element diffracting crystals (LDEB designed for boron analysis, and LDE1 for carbon), in differential mode, to avoid possible interferences. Special care was paid for boron analysis, as it is present as a trace element in the sample (<1 wt.%), and it holds several analytical challenges (broad peak, peak shifts easily depending on the type of molecule in which boron is engaged, severe secondary fluorescence, strong X-ray absorption, high-diffusivity during heating, potential overlap with other elements). In order to identify the peak of boron, a peak search was repeatedly performed in both standards specimen (BN) and the sample, in different conditions. The peak was identified at L-value of 129.70 mm, which represents a significant peak shift compared to the boron peak in compounds with nitrogen or oxygen (~126.80 mm). Since the peak of boron is very wide (also specific for other light elements), large offset backgrounds were selected (lower offset 22 mm, upper offset 12 mm). Several sets of accelerating voltage and beam current were used for precisely identify the kα peak position and optimal X-ray intensity of boron and carbon. Accelerating voltage higher than 15 keV and beam current higher than 20 nA do not provide a good signal for boron (x-ray intensities were below detection limit), due to the loss of boron during the beam exposure caused by the high thermal diffusivity of B). Accelerating voltage lower than 15 keV and beam current lower than 20 nA do not produce enough x-

rays of boron, as boron has trace concentration in the samples. The optimal analytical conditions employed for quantitative analysis were: 15 keV acceleration voltage and 20 nA beam current. As the sample is beam-sensitive, a defocused 20-microns beam was used in order to limit the sample damaging and the loss of boron. To further limit the volatility of boron during heating under the electron beam, and with the risk of increasing the detection limit of boron, a low counting time was used (8 s per peak and 4 s for each, lower and upper background, respectively). The counting time of the X-rays for carbon (a major element in the sample) was 10 s for peak and 5 s for each lower and upper background, respectively. The PRZ (JEOL) matrix correction was employed for quantification. Standards used for quantification were boron nitride (BN) for boron and graphite (C) for carbon. The average detection limit (DL) for boron is $569 \pm 27$ ppm, while the average analytical standard deviation (error of each analysis based on x-ray intensities of the peak and background offsets and counting times) is $12.04 \pm 1.2\%$. WDS quantitative elemental maps were acquired at 15 keV accelerating voltage and 20 nA beam current, using stage mode with 10 ms dwell time. The same standards used for quantitative analysis were employed for the quantification of the element maps. Deadtime correction was applied for each element map.

**DFT calculations and AIMD simulations**. All the DFT calculations and AIMD simulations are conducted with the VASP package (V5.4.4) where the solution model VASPsol module is included. All the structure relaxation and molecular dynamics were carried out based on spin-polarized DFT using the Perdew–Burke–Ernzerhof functional and plane-wave basis sets of 400 eV cutoff kinetic energy. The geometry relaxation and the electronic self-consistent field convergence criterion were set to 0.01 eV/Å and $10^{-5}$ eV, respectively. The dispersion correction counted by DFT-D3 method. $4 \times 4$ graphene supercell with 25 Å of vacuum space to model the doped C materials, for which $3 \times 3 \times 1$ Monkhorst-Pack sampled k-points are used. To consider the charge effects, constant-potential ($\mu_e$) method was adopted based on the Vaspsol implicit solvation model[49].

The free energy of each state is referred to $O_2(g)$, $H_2(g)$ and pure surface as the following:

$$\Delta G(^*O_xH_y) = E(^*O_xH_y) - E(^*) - x/2^*E(O_2(g)) - y/2^*E(H_2(g)) - (n2 - n1)\mu_e + Corr. \quad (3)$$

Where the two terms are the total energy of the surface with and without the $O_xH_y$ adsorbate, the third and fourth terms corresponds to total energy of $O_2$ and $H_2$ reference, the fifth term is the charging energy where $n2$ and $n1$ is the net charge (unit: $e^-$) of the surface with and without adsorbate and $\mu_e$ refers to the electron energy in RHE (i.e., $U_{RHE} = \mu_e - \mu_{SHE} + 0.059 \times pH$). The last term is the sum of all zero-point energy, enthalpy, and entropy terms of the surface and gas-phase species (see Supplementary Table 8).

In order to evaluate the kinetic barriers, we employed a constrained MD method, the "slow-growth" approach to obtain the free-energy profile[50]. The charge is adjusted on-the-fly to keep the potential to be constant. In this method, the value of the reaction coordinate (namely $\xi$) is linearly changed from the characteristic value for the initial state (IS) to that for the final state (FS) with a velocity of transformation $\dot{\xi}$. The resulting work needed to perform a transformation from IS to FS can be computed as:

$$w_{IS \to FS} = \int_{\xi(IS)}^{\xi(FS)} \left(\frac{\partial F}{\partial \xi}\right) \cdot \dot{\xi} dt, \quad (4)$$

where $F$ is the free-energy calculated at general coordinate $q$ which is evolving with $t$, $\frac{\partial F}{\partial \xi}$ is calculated along the track of a constrained MD through the SHAKE algorithm[51]. In the limit of infinitesimally small $\partial \xi$, the work $w_{IS \to FS}$ corresponds to the free-energy difference between the final and initial state.

**Electrochemical cell performance test**. For the three-electrode flow-cell setup, all potentials measured against SCE was converted to the RHE scale in this work using

$$E \, vs. RHE = E \, vs. SCE + 0.241 \, V + 0.0591 \times pH \quad (5)$$

where pH values of electrolytes were determined by Orion 320 PerpHecT LogR Meter (Thermo Scientific). 1 M KOH (pH = 14) and 1M $Na_2SO_4$ (pH = 7) were used as the aqueous electrolyte. The flowrate for the aqueous electrolyte was fixed at 54 mL h$^{-1}$ controlled by a peristaltic pump. Solution resistance (Rs) was determined by potentiostatic electrochemicalimpedance spectroscopy at frequencies ranging from 0.1 Hz to 200 kHz. All the measured potentials using three-electrode setup were manually 100% $iR$-compensated. The cathode electrode was prepared by uniformly spray coating the well-mixed catalyst ink on Sigracet 39 BC GDL electrode with the loading of B–C around 0.5 mg cm$^{-2}$. The ink is prepared by adding B–C powders and mixed with 4 µL mg$^{-1}$ binder (nafion 117), together with enough isopropanol as the solvent, and sonicate for 30 min until the solution is uniformly dispersed.

For the two-electrode solid-electrolyte cell for electrosynthesis of pure $H_2O_2$, an AEM (Dioxide Materials and Membranes International Inc.) and a Nafion 117 film (Fuel Cell Store) were used for anion and cation exchange, respectively. Around 0.5 mg cm$^{-2}$ B–C and $IrO_2$ loaded on Sigracet 39 BC GDL electrode (about 4 cm$^2$ electrode area) were used as cathode and anode, respectively. The cathode side was supplied with 20 sccm of $O_2$ gas feed rate. The anode side was cycled with 1 M $H_2SO_4$ aqueous solution at 162 mL h$^{-1}$. The measured potentials using two-

electrode solid-electrolyte cell setup were manually 100% $iR$-compensated afterwards.

The generated $H_2O_2$ concentration $C_{titration}$ was evaluated using standard potassium permanganate (0.1 N $KMnO_4$ solution, Sigma-Aldrich) titration process, according to the following equation:

$$5H_2O_2 + 2KMnO_4 + 3H_2SO_4 \to 5O_2 + 2MnSO_4 + K_2SO_4 + 8H_2O \quad (6)$$

Theoretical concentration that the cell can generate if the FE is 100% is given by:

$$C_{theory} = \frac{I}{2 \times 96485 \times L} \quad (7)$$

Where $I$ is the current applied and L is the electrolyte/DI water flow rate in the cell system. Then the selectivity of $H_2O_2$ can be calculated as:

$$Faradaic \, efficiency \, of \, H_2O_2(\%) = 100 \times \frac{C_{titration}}{C_{theory}} \quad (8)$$

## Data availability
The data that support the plots within this paper and other finding of this study are available from the corresponding authors upon reasonable request.

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

## Acknowledgements

This work was supported by Rice University. H.W. is a CIFAR Azrieli Global Scholar in the Bio-inspired Solar Energy Program. The use of the EPMA facility at the Department of Earth, Environment and Planetary Science, Rice University, Houston, TX, is kindly acknowledged. The DFT study is supported by the NSF (1900039), the Welch Foundation (F-1959-20180324), and ACS-PRF (60934-DNI6). The DFT calculations use the computational resources provided by National Renewable Energy Lab, the XSEDE (TG-CHE190065), the Argonne National Lab, and the Brookhaven National Lab. We thank Dr. Gelu Costin for his assistance and guidance during the electron microprobe analysis. C.X. thanks the support from J. Evans Attwell-Welch Postdoctoral Fellowship.

## Author contributions

Y.X., C.X., and H.W. conceptualized the project. H.W. and Y.L. supervised the project. Y.X. developed and performed catalyst synthesis and conducted the catalytic tests of catalysts and the related data processing. Y.X. performed materials characterization with the help of P.Z., J.Y.K., and G.G. XAS experiments were carried out by J.Z. and Y.H. during the revision of the manuscript. X.Z. and X.B. performed the DFT simulation. Y.X., X.Z., Y.L., and H.W. wrote the paper. C.X. and Z.Y.W. helped the revision of the paper. All authors discussed the results and commented on the paper.

## Competing interests

The authors declare no competing interests.
