## [Peer Review File · Nature Communications]

Reviewer #1 (Remarks to the Author):

This manuscript covers organized experimental works on elucidating the role of boron dopant on carbon materials to achieve industrial-relevant H₂O₂ production rates. Though the beneficial impact of boron-doped carbon support is well-demonstrated and the authors made inspiring experimental approaches and performance with their porous solid electrolyte reactors, a set of the experimental results in the manuscript does not show a strong correlation between the performance and boron sites. Our opinions are that the authors couldn't fully exclude the effect of co-doped oxygen dopant site, which has been recognized as promising active site for electrocatalytic H₂O₂ production recently. Therefore, this manuscript is not suitable enough to be published in Nature Communications for the following reasons;

1. The synthesized boron-doped carbon (B-C) has only 0.41 at.% on their catalyst surface, but the oxygen content of B-C is even 1.25 at.%. The authors argued that these oxygen functional groups are the trace amount of oxygen from the adsorption of water species or air gas. However, there is short of evidence to support that argument, because the most carbonized carbon materials at high temperature (over 750 °C) also have a certain amount of oxygen species on their carbon surfaces [ref: "Nat Catal (2020). In press, doi.org/10.1038/s41929-020-00546-1; J. Am. Chem. Soc. 2017, 139, 14143-14149; J. Am. Chem. Soc. 2020, 142, 2404-2412]. The authors conducted XPS analysis to investigate the oxygen species, but it would be better to obtain NEXAFS O K-edge spectra as reported in previous reports [ref: Sci. Rep. 2019, 9, 1532.], which can confirm that the oxygen content was contributed from the water trace. This is the key point of this manuscript, so it should be more thoroughly discussed.
2. In this study, the synthesized doped carbons have certain amount of each dopant site. The experiment suggested the atomic ratio of B-C, N-C, P-C and S-C catalysts is 0.41, 0.86, 0.34, and 0.90 at.% for boron, nitrogen, phosphorous and sulfur dopant. More precisely synthetic adjustment to make quite similar contents of dopant species would be required to eliminate the possibility of different amount of ORR active sites.
3. Stability- There is no significant morphological change after the reaction with TEM, but the chemical configuration change of the catalyst surface after the stability test is not proven (XPS or Raman... etc.). There is a possibility that B-C is more likely to be easily oxidized in ORR reaction process than other dopant cases, which can mean boron sites are not direct active sites for 2e⁻ ORR, but well convert B-C to oxidized carbons.
4. One of the interesting results in the manuscript is the adverse effect of oxidized carbon electrode with sluggish ORR kinetics especially under high currents. The authors suggested that this could be stem from their high charge transfer resistivity. However, in Fig. S21 and S22, the ECSA of O-C is much larger than that of pure C or B-C. More in-depth discussion is necessary to understand the ORR kinetics and charge transfer process on the carbon surface.
5. The ORR performance of O-C in RRDE setup should be also shown in Fig. 1 under low current ranges.
6. The BET surface area of O-C is omitted.

Reviewer #2 (Remarks to the Author):

Xia et al. presented a synthesis and characterization of differently doped carbon and tested them for O₂ reduction to hydrogen peroxide.

Even though their work is intensive and scientifically correct, overall results are not suitable to Nat. Commun. Synthesis of boron or nitrogen doped carbon is not new at all and their capability to reduce oxygen to H₂O₂ is not superior or interesting compared to previous results. Overall, it is a good piece of work in itself and I won't be surprised to see it appearing in places like Electrochimica Acta. But it lacks the level of excitement in terms of conceptual breakthrough or technical advancement for me to recommend it for publication here.

Reviewer #3 (Remarks to the Author):

This work reported a boron-doped carbon (B-C) catalyst which presented enhanced activity under high currents (up to 300 mA cm⁻²) while maintaining high H₂O₂ selectivity (~ 90%). However, there are still many problems to be clarified and some experiments are lack of. A substantial modifications are needed.

1. Page 6 - "non-metal elements with atomic ratio of 0.41, 0.86, 0.34, and 0.90 at.% for boron, nitrogen, phosphorous and sulfur dopant, respectively". So that the content of heteroatom in the catalysts was different. It is difficult to compare the performance since the catalyst loading is different.
2. Is it the characteristics of heteroatom or the content of heteroatom resulted in the difference of ORR activity? Please eliminate the influence of content of heteroatom content.
3. As shown in Fig. 1, the selectivity of H₂O₂ by P-C was higher than other four catalysts although the onset potentials was lower than B-C. However, the author's conclusion was B-C was best. Therefore, what is the decisive factor? And the how to choose this factor?
4. Even if B-C was the best, have they investigated the effects of B doping amount?
5. Some more details information on experiments are missing. For example, how the cathode was prepared using the B-C catalyst.
6. In DFT calculations, the graphene was used as carbon structure in configurations of heteroatom doping carbon black, is this appropriate?
7. In Fig. 5b and 5c, the Higher H₂O₂ activities by B-C was showed compared with O-C, however, the FE was lower. Why?
8. Page 17 - "With the incorporation of our B-C catalyst into this cell configuration as the anode, under a fixed DI water feed rate" Should the B-C catalyst be used as cathode?
9. For prepared B-C, why did the remnant need to be annealed, however, Other catalysts are not needed?
10. In Fig. 5g, it is suggested to provide generation of H₂O₂ at different pH conditions.
11. In Table S4, it would be more convinced to compare more non-metal doped carbon materials.
12. For industrial application, 25h stability test is not sufficient.
13. In solid electrolyte cell, what effect if not using 1 M H₂SO₄?
14. It is suggest to give energy consumption analysis or H₂O₂ generation cost to check the advantage of this method.

Title: “**Highly Active and Selective Oxygen Reduction to H₂O₂ on Boron-Doped Carbon for High Production Rates**”

Authors: Yang Xia, Xunhua Zhao, Chuan Xia, Zhen-Yu Wu, Peng Zhu, Jung Yoon (Timothy) Kim, Xiaowan Bai, Guanhui Gao, Yongfeng Hu, Jun Zhong, Yuanyue Liu, Haotian Wang

Corresponding authors: Yuanyue Liu, Haotian Wang

We thank the reviewers for the constructive comments which have helped us to greatly improve our research and the quality of our manuscript. We have now included additional analysis and characterizations and performed substantial experiments to fully address the reviewers’ concerns and suggestions. **Furthermore, we have modified our manuscript and SI based on the additional results and analysis, which are highlighted in yellow.** Below, we address the points raised by reviewers one by one.

Reviewer 1

This manuscript covers organized experimental works on elucidating the role of boron dopant on carbon materials to achieve industrial-relevant H₂O₂ production rates. Though the beneficial impact of boron-doped carbon support is well-demonstrated and the authors made inspiring experimental approaches and performance with their porous solid electrolyte reactors, a set of the experimental results in the manuscript does not show a strong correlation between the performance and boron sites. Our opinions are that the authors couldn’t fully exclude the effect of co-doped oxygen dopant site, which has been recognized as promising active site for electrocatalytic H₂O₂ production recently. Therefore, this manuscript is not suitable enough to be published in Nature Communications for the following reason.

Response #1: Thank you for your important comments here. We appreciate for your positive feedback on the performance of our B-C catalyst and our experimental approaches. We

completely understand the reviewer's major concern on the oxygen effects. To further exclude the possible effect of oxygen species, we have now included additional analysis and characterizations and performed substantial experiments. All the experimental evidence can strongly support that the excellent performance of B-C originates from B dopants instead of O dopants (See our point-to-point response below). Thus, we believe that with the amendment of additional experiments and characterizations that we have done, we can now successfully resolve the reviewer's concern and also dramatically improve the quality of our work to meet the high standards of *Nature Communications*. Specifically, we have addressed your questions and concerns as follows:

1. The synthesized boron-doped carbon (B-C) has only 0.41 at.% on their catalyst surface, but the oxygen content of B-C is even 1.25 at.%. The authors argued that these oxygen functional groups are the trace amount of oxygen from the adsorption of water species or air gas. However, there is short of evidence to support that argument, because the most carbonized carbon materials at high temperature (over 750 °C) also have a certain amount of oxygen species on their carbon surfaces [ref: "Nat Catal (2020). In press, doi.org/10.1038/s41929-020-00546-1; J. Am. Chem. Soc. 2017, 139, 14143-14149; J. Am. Chem. Soc. 2020, 142, 2404-2412]. The authors conducted XPS analysis to investigate the oxygen species, but it would be better to obtain NEXAFS O K-edge spectra as reported in previous reports [ref: Sci. Rep. 2019, 9, 1532.], which can confirm that the oxygen content was contributed from the water trace. This is the key point of this manuscript, so it should be more thoroughly discussed.

Response #1.1: We highly appreciate the reviewer's important comments and suggestion. This is a really good question. We do agree that XAS technique is a suitable way to analyze this problem. The papers that the reviewer referred here are good examples that show the use of this technique. Thus, we have performed the soft X-ray spectroscopy for O K-edge spectra for all the doped carbon samples. The spectrum data are shown below as Figure R1 and also included in the revised SI (Figure S16). For all of the four doped materials, there are three peaks in the range of 520-560 eV. The 1st peak around 533 eV for all of the samples is assigned to the π -bonded oxygen, which might come from possible trace amount of oxygen dopant which is impossible to remove when annealing the samples at the high temperature (750 °C) as Reviewer 1 suggested. It might also be formed especially on some active sites with dangling bonds once the sample is exposed to air. This is also the reason why we completely agree with

the reviewer as well as those recommended references that it is unlikely to completely get rid of oxygen in carbon materials. However, the other two peaks, which dominate the O K-edge signal at around 535 eV and 540 eV, can be assigned to residual/adsorbed water during sample exposure to air (Chem. Rev. 2020, 120, 4056–4110). Given the data from XAS, together with the XPS results that we have done (Figure R2), we believe that it is safe to conclude that the surface oxygen detected by our XPS analysis is mainly from the adsorbed/residual water species, with a small portion of possible remaining oxygen coming from O dopants or oxidation during the sample exposure to air. We have also attached Figure S15 here for the O high resolution peaks for all the doped materials.

Figure R1. NEXAFS O K-edge spectra of B-C, N-C, P-C, and S-C. The peak located at ~533 eV for all the samples corresponds to π -bonded oxygen on carbon surface either from trace amount of oxygen dopant which is impossible to remove when annealing the samples at the high temperature (750 °C) suggested by Reviewer 1 or due to the oxidation during the sample exposure to air. The peaks around 535 eV and 540 eV are assigned to surface adsorbed/residual water species, which is consistent with our XPS results shown below and in SI.

Figure R2. High resolution oxygen XPS peak analysis of (a) B-C (b) N-C (c) P-C (d) S-C (e) Pure C, respectively. The peak located at 532~533 eV for all the five samples corresponds to surface adsorbed $\text{H}_2\text{O}/\text{OH}^-$ species. Since there is no peak of lattice oxygen (around 529~530 eV (Chemical Society Reviews, **46** (10), 2017)) observed, we can conclude that the trace amount of oxygen species remained come from the adsorption of water and air during XPS sample preparation and transportation.

On the other hand, additional control experiments further exclude the impact of oxygen species on B-C catalyst's performance. While a precise control of oxygen percentage in different samples is impossible, we tried our best to prepare a control sample (B-C-36) that has higher doping level of B while lower doping level of O (see Table S6 and also Table R1 below). Compared to the original B-C catalyst, the B-C-36 synthesized from more activated carbon substrate shows roughly double amount of boron dopants while less amount of oxygen. Detailed synthetic method is included in our Methods section. Specifically, B increases to 0.91

at% from 0.41 at%, and O decreases to 1.25 at% to 0.85 at%. As a result, the B-C-36 catalyst showed significantly improved $2e^-$ -ORR activity compared to our original B-C catalyst while maintaining similar or even slightly better H_2O_2 FEs (see Figure R3 and Figure S31). This performance improvement can now be safely ascribed to the increased density of B dopants as the oxygen doping level is actually decreased. A more direct comparison is between Pure C and B-C-36, where the oxygen level decreased from 0.99 at% to 0.85 at%. Obviously, with the B dopants and lower oxygen level, B-C-36 presented dramatically improved H_2O_2 activity (Figure R3). This significant improvement cannot be explained by the lower contents of oxygen dopants. Based on these additional experiments, we do believe that it is the boron dopant that plays the central role in enhancing the $2e^-$ -ORR performance to produce H_2O_2 , rather than the trace amount of lattice oxygen. The above discussions have also been included into revised SI (Page 17).

Table R1. Atomic ratios of Pure C, B-C and B-C-36 determined by XPS.

Atomic Ratio	C (at. %)	B (at. %)	O (at. %)
Pure C	99.01	0	0.99
B-C	98.34	0.41	1.25
B-C-36	98.24	0.91	0.85

Figure R3. Three-electrode flow cell performance of Pure C, B-C and B-C-36 in 1M Na₂SO₄. (a) I-V curve and (b) corresponding faradaic efficiencies. Note that all the I-V curves and faradaic efficiency were taken average of 2 independent tests for each of the samples. All the I-V curves are manually iR -compensated.

2. In this study, the synthesized doped carbons have certain amount of each dopant site. The experiment suggested the atomic ratio of B-C, N-C, P-C and S-C catalysts is 0.41, 0.86, 0.34, and 0.90 at.% for boron, nitrogen, phosphorous and sulfur dopant. More precisely synthetic adjustment to make quite similar contents of dopant species would be required to eliminate the possibility of different amount of ORR active sites.

Response #1.2: Thank you for your constructive comment here. We do agree that there is a difference in the doping level that we got for our series samples because we used a general and very facile method for the synthesis and the amount of the precursor is excessive. In fact, it is extremely difficult and unlikely to make all the dopants at exactly the same doping level in the carbon lattice. However, even without the same doping levels, it is still straightforward for us to make comparisons and convincing conclusions. As shown in Figure 1, the RRDE analysis exhibits that even some of the dopants have slightly higher ratio, e.g. nitrogen in N-C and sulphur in S-C, the performance of those two samples are even worse than B-C with a lower dopant loading. Thus, it is believed that the difference in dopant levels does not play a direct and important role here. Furthermore, we have slightly modified our synthetic procedures to introduce more dopants (B, N, P, and S) into carbon black while keeping the content of O at nearly the same level as previous samples. Detailed synthetic method is included in our Methods section. We have included the atomic ratio for each of the new samples in Table R2 and also in SI Table S7. In terms of the performance, the trend for all the doped samples are the same compared to the RRDE performance (Figure R4). Specifically, B-C-36 has the best kinetics and onset potential among all the four dopants, with N-C-36 having the 2nd best kinetics but lowest FE towards H₂O₂ production, followed by P-C-36 and S-C-36 with similar kinetics. A more convincing comparison can be made between our previous B-C (0.41% of B) and newly synthesized N-C-36 (1.12%), S-C-36 (1.06%) and P-C-36 (0.7%), where its B doping level is lower than all of other dopants but its 2e⁻-ORR performance is much higher than all other high doping level catalysts (Figure R4). Therefore, we can safely conclude that it is the type of dopant species, not their concentrations, that play the most important role in determining the ORR to H₂O₂ performances. Figure R4 has also been included into revised SI (Figure S32).

Table R2. Atomic ratios of X-C-36 (X represents B, N, P and S) series catalysts determined by XPS.

Atomic Ratio	C (at. %)	Heteroatom (at. %)	O (at. %)
B-C-36	98.24	0.91	0.85
N-C-36	98.00	1.12	0.88
P-C-36	98.38	0.70	0.92
S-C-36	98.42	1.06	0.52

Figure R4. Three-electrode flow cell performance of catalysts. (a) I-V curve for B-C, B-C-36, N-C-36, P-C-36 and S-C-36 in 1 M Na₂SO₄ (b) Corresponding faradaic efficiencies measured. All the I-V curves are manually iR -compensated. B-C-36 shows highest selectivity among all the dopants, with the best activity as well. Furthermore, B-C with lower B loading still shows superior ORR performance compared to other dopants in the X-C-36 series with higher dopant level.

3. Stability- There is no significant morphological change after the reaction with TEM, but the chemical configuration change of the catalyst surface after the stability test is not proven (XPS or Raman... etc.). There is a possibility that B-C is more likely to be easily oxidized in ORR reaction process than other dopant cases, which can mean boron sites are not direct active sites for 2e- ORR, but well convert B-C to oxidized carbons.

Response #1.3: Thank you for your important comment here. According to your suggestion, we performed the XPS of B-C after the catalysis test. There is no obvious increase in the oxygen content, nor obvious decrease in boron content (Table R3). Furthermore, there is no

significant shift in the boron peak, indicating the boron oxidation state has been well kept during the reaction (Figure R5). Thus, we can safely conclude that it is the boron, instead of other oxidized boron species, plays the role in the enhancing performance for $2e^-$ ORR towards H_2O_2 production. Moreover, more oxidized carbons might give higher selectivity, but not necessarily giving better kinetics, which is exactly demonstrated in this work. The data below have been included into revised SI (Table S5 and Figure S27).

Table R3. Atomic ratios of B-C before and after ORR determined by XPS.

Atomic Ratio	C (at. %)	B (at. %)	O (at. %)
B-C before reaction	98.34	0.41	1.25
B-C after reaction	98.5	0.45	1.05

Figure R5. XPS High resolution B peak analysis before and after ORR. (a) before ORR (b) after ORR. The oxidation state of boron after the reaction has been well kept, and the boron atomic ratio kept similar, indicating the boron species has not been oxidized during ORR.

4. One of the interesting results in the manuscript is the adverse effect of oxidized carbon electrode with sluggish ORR kinetics especially under high currents. The authors suggested that this could be stem from their high charge transfer resistivity. However, in Fig. S21 and S22, the ECSA of O-C is much larger than that of pure C or B-C. More in-depth discussion is necessary to understand the ORR kinetics and charge transfer process on the carbon surface.

Response #1.4: Thank you very much for your important comment here. ECSA is estimated from the electrochemical double layer capacitance of the catalyst surface. It only depends on

the number of active sites exposed on the catalysis reaction interface, but it has nothing to do with the activity per active site. Thus, it is possible that a material with a higher ECSA exhibits lower activity than a second material with a lower ECSA. An intuitive example can be the comparison of HER activities between graphene oxide nanosheets and Pt/C, where graphene oxide nanosheets may present several times higher ECSA than Pt/C but showed extremely low HER activity. On the other hand, charge transfer resistance indicates the ability for charge transfer. It is a totally different concept from either the double layer capacitance or ECSA. As an example, a typical single RC circuit is a parallel of a double layer capacitance and a charge transfer resistance. High ECSA does not necessarily leading to low charge transfer resistance. For O-C, there is a much higher oxygen species on the catalyst surface, which means the number of exposed active sites should be much higher and thus makes the onset for O-C more positive. Thus, it is reasonable that the ECSA for O-C is much higher than that of pure C and B-C. However, the oxidized species in carbon materials will typically lead to poor electrical conductivity as well as slow charge transfer, as mentioned in the manuscript (Page 4). When the current becomes larger, the ability for charge transfer becomes the rate-limiting step, which makes the kinetics of O-C sluggish.

5. The ORR performance of O-C in RRDE setup should be also shown in Fig. 1 under low current ranges.

Response #1.5: Thank you for your nice comment here. We have added the RRDE performance of O-C in 0.1 M KOH in the supplementary section for reference because the performance of O-C in RRDE is not directly related to our topic in this work. O-C on RRDE has been demonstrated before having a good performance. However, its performance under low current densities cannot reflect its sluggish kinetics under high current regions. Please see Figure. R6 for the ORR performance of O-C on RRDE setup (also included in SI Figure S6). Note that even though the performance is good, which is expected for oxidized carbon under low current ranges, its sluggish kinetics under high current ranges are the area that we would like to focus.

Figure R6. ORR performance of O-C by RRDE 0.1M KOH. (a) Linear sweep voltammetry (LSV) (b) H₂O₂ molar selectivity.

6. The BET surface area of O-C is omitted.

Response #1.6: Thank you for your invaluable comments. Please see Figure. R7 for the BET of O-C. We can see that the BET surface area of O-C is 213.6 m² g⁻¹. This figure has also been included into our revised SI (Figure S14).

Figure R7. BET surface area analysis for O-C. Note that the BET surface area of O-C is slightly higher than B-C (149.5 m² g⁻¹), excluding any possible contribution of the surface area increase to the enhanced H₂O₂ performance of B-C.

Reviewer 2

Xia et al. presented a synthesis and characterization of differently doped carbon and tested them for O₂ reduction to hydrogen peroxide. Even though their work is intensive and scientifically correct, overall results are not suitable to Nat. Commun. Synthesis of boron or nitrogen doped carbon is not new at all and their capability to reduce oxygen to H₂O₂ is not superior or interesting compared to previous results. Overall, it is a good piece of work in itself and I won't be surprised to see it appearing in places like Electrochimica Acta. But it lacks the level of excitement in terms of conceptual breakthrough or technical advancement for me to recommend it for publication here.

Response #2: Thank you very much for your valuable comments on our manuscript. First, we completely agree with the reviewer that the synthesis of boron or nitrogen doped carbons have been widely reported before. But we want to point it out that our work is not focused on the synthetic part but the use of B-C for industrial-relevant 2e⁻-ORR to H₂O₂ performance. Second, the reviewer mentioned that our B-C's H₂O₂ activity is not superior compared to previous results. We however respectively disagree with this point. To our best knowledge, the performance of our B-C both in activity and stability, especially under industrial-relevant production rates as the central focus of this work, presents a dramatical improvement compared to the-state-of-the-art catalysts reported before (see Table S4 for the comparison of the performance of our B-C to recently reported 2e⁻-ORR catalysts). More importantly, it is also for the first time to demonstrate a continuous generation of pure H₂O₂ solutions under such large H₂O₂ current densities in a solid electrolyte reactor. We believe those industrial-relevant performances represent a new benchmark in the field and well demonstrate the great promise of large-scale applications in the future. We have also attached the table as Table R4 below.

Table R4. Comparison of electrochemical O₂-to-H₂O₂ performance under high currents with state-of-the-art catalysts.

Catalyst	Electrolyte	Potential	j_{total} (mA cm ⁻²)	FE (%)	Stability
This work	1M KOH	0.685 V vs. RHE	300	85.1	200 hrs @ 30 mA cm⁻²
	1M Na₂SO₄	0.277 V vs. RHE	300	83.2	
	DI water (Solid-electrolyte reactor)	2.55 V	400	85.5	
Fe-O-CNT ³	1M KOH	0.76 V vs. RHE	45	95.4	8 hrs @ 1-3 mA cm ⁻²
Natural air diffusion electrode ⁴	0.05 M Na ₂ SO ₄	Not mentioned	240	66.8	20 hrs @ 60mA cm ⁻²
O-CNT ⁵	1M KOH	0.68 V vs. RHE	40	90	10 hrs @ 0.2-0.4 mA
Ni-N ₂ O ₂ /C ⁶	0.1M KOH	0.5 V vs. RHE (without iR -compensation)	70	91	8 hrs @ 70 mA cm ⁻²
Co-N-C ⁷	0.1M KOH	~0.55 V vs. RHE	50	Not mentioned	110 hrs @ ~2.4 mA cm ⁻²
C-PTFE electrode ⁸	0.05M Na ₂ SO ₄	Not mentioned	145	29	N/A
PtP ₂ -Al ₂ O ₃ ⁹	membrane fuel cell (H ₂ anode)	Not mentioned	150	78.8	110 hrs @ 0.4V (current density not mention)
N-doped mesoporous carbon ¹⁰	0.1M KOH	0.3 V vs. RHE	2 in RRDE	82	6 hrs @ 3 mA cm ⁻²
	0.1M K ₂ SO ₄	0.2 V vs. RHE	3 in RRDE	75	6 hrs @ 3.5 mA cm ⁻²
B,N doped carbon ¹¹	0.1M KOH	0.55 V vs. RHE	1.5 in RRDE	85	50 hrs @ ~1.2 mA cm ⁻²
N-doped carbon nanohorns ¹²	0.1M NaOH	0.65 V vs. RHE	~0.6 in RRDE	~65	25 hrs @ ~0.6 mA cm ⁻²
	0.1M PBS	0.45 V vs. RHE	~0.5 in RRDE	~90	25 hrs @ ~0.5 mA cm ⁻²

3 Jiang, K. *et al.* Highly selective oxygen reduction to hydrogen peroxide on transition metal

- single atom coordination. *Nature communications* **10**, 1-11 (2019).
- 4 Zhang, Q. *et al.* Highly efficient electrosynthesis of hydrogen peroxide on a superhydrophobic three-phase interface by natural air diffusion. *Nature communications* **11**, 1-11 (2020).
- 5 Lu, Z. *et al.* High-efficiency oxygen reduction to hydrogen peroxide catalysed by oxidized carbon materials. *Nature Catalysis* **1**, 156-162 (2018).
- 6 Wang, Y. *et al.* High-Efficiency Oxygen Reduction to Hydrogen Peroxide Catalyzed by Nickel Single-Atom Catalysts with Tetradentate N₂O₂ Coordination in a Three-Phase Flow Cell. *Angewandte Chemie International Edition* (2020).
- 7 Jung, E. *et al.* Atomic-level tuning of Co–N–C catalyst for high-performance electrochemical H₂O₂ production. *Nature Materials* **19**, 436-442 (2020).
- 8 Brillas, E., Calpe, J. C. & Casado, J. Mineralization of 2, 4-D by advanced electrochemical oxidation processes. *Water Research* **34**, 2253-2262 (2000).
- 9 Li, H. *et al.* Scalable neutral H₂O₂ electrosynthesis by platinum diphosphide nanocrystals by regulating oxygen reduction reaction pathways. *Nature communications* **11**, 1-12 (2020).
- 10 Sun, Y. *et al.* Efficient Electrochemical Hydrogen Peroxide Production from Molecular Oxygen on Nitrogen-Doped Mesoporous Carbon Catalysts. *ACS Catalysis* **8**, 2844-2856, doi:10.1021/acscatal.7b03464 (2018).
- 11 Chen, S. *et al.* Designing boron nitride islands in carbon materials for efficient electrochemical synthesis of hydrogen peroxide. *Journal of the American Chemical Society* **140**, 7851-7859 (2018).
- 12 Iglesias, D. *et al.* N-doped graphitized carbon nanohorns as a forefront electrocatalyst in highly selective O₂ reduction to H₂O₂. *Chem* **4**, 106-123 (2018).

We can see clearly the superior performance of our B-C catalyst compared to other reported catalytic materials. More importantly, in our work, we have performed a systematic study on how different dopant will affect the 2e⁻-ORR performance of carbon materials, and found great activity of B-C in large current H₂O₂ generation (up to 400 mA cm⁻²) and great stability of over 200 hours. To demonstrate additional novelty of our work, we have also made detailed discussion below. (The novelty of this work has already been included in our manuscript on Page 3-5 and in the discussion part.)

- 1. The 2e⁻-ORR performances reported so far are mostly focused in RRDE setup and are limited within a few mA cm⁻² due to O₂ mass diffusions. These studies are highly important for fundamental studies, but the current density range is far away from potential industrial large-scale applications. In our study, by designing**

our B-C active catalyst and using our solid electrolyte reactor, we successfully achieved high current densities (up to 400 mA cm⁻²), high H₂O₂ selectivity (~ 90%) and high stability (200 hours continuous operation), demonstrating a great promise of B-C catalyst to be used for future practical applications.

Several recent studies showed some dopant effects (such as O, Fe, *etc.*) on carbon in 2e⁻-ORR, but they are typically focused on onset potential regions with low ORR current densities, which is only the first step for screening a catalyst's capability in producing H₂O₂ via ORR. **Those cases are different from the industrial-relevant performance scenarios.** Thus, it is still urgent and important for us to find out a promising catalyst that can help to maintain high H₂O₂ selectivity while delivering significant ORR current densities under small overpotentials. The stability demonstrated before is typically within tens of hours, which is also far away from the potential industrial practical applications. In this work, on the other hand, we have shown the outstanding performance of B-C catalyst under large currents (up to 400 mA cm⁻²) with superior stability of at least 200 hours, which helps to bridge the gap between the lab-scale catalyst design and future H₂O₂ practical applications.

2. There is no systematic study on the effect of different non-metal doping on 2e⁻-ORR.

Non-metal dopants in carbon materials have been known to be able to serve as the active sites in catalysis or impact the electronic properties of surrounding carbon atoms by being incorporated into the carbon lattice. There are a few non-metal doping carbon materials reported in recent years capable of producing H₂O₂ by ORR. However, they only focused on one single dopant (*ACS Catalysis*, **8**, 2844-2856 (2018)) (*Chem*, **4** 106-123 (2018)) or mixed dopant effect like boron nitride (*JACS*, **140(25)**, 7851–7859 (2018)) on different carbon platforms (including carbon black, CNT, GO, mesoporous carbon), which makes it difficult to give a direct indication of which non-metal dopant is more favorable in enhancing H₂O₂ production due to the difference in the carbon substrate. In this work, we have synthesized a series of non-metal doping materials **on the same carbon platform**, so that we can directly compare different doping effect without adding the factors of background effect of different carbon substrates.

3. There has been no systematic DFT calculations to reveal the effect of different non-metal dopants on the 2e⁻-ORR performance.

Previous studies have some simple calculations for a single active site but lacks a systematic study on different doping sites. Moreover, those studies are more focused on thermodynamics aspect, without too much emphasis on the charge effect on the reaction intermediate. In this work, density functional theory (DFT) was used to calculate the adsorption free energy of key intermediates in ORR and conducted ab-initio molecular dynamics at constant potential to model the reaction conditions. It is revealed that B doped at single vacancy has nearly-zero overpotential, while molecule dynamics at constant potential indicates that the energy barrier for 2e⁻ pathway is lower than its 4e⁻ counterpart. Thus, B dopants were finally identified the best for 2e⁻ pathway for the first time.

Thus, we do believe that there is strong technical advancement and novelty in this work. The focus of this work is not on the material synthesis, but rather on how the kinetics are improved while not sacrificing the H₂O₂ selectivity on carbon materials for practical applications. We have done a systematic study on how different non-metal dopants will affect H₂O₂ production via ORR on the same carbon platform. We have improved the overpotential at industrial-relevant level currents by hundreds of millivolts, which will eventually save a significant amount of energy. The good stability of at least 200 hours has also been demonstrated, which is highly important in terms of its potential in the practical applications. We hope the above arguments can help to resolve the reviewer's concern on the catalytic performance of our work.

Reviewer 3

This work reported a boron-doped carbon (B-C) catalyst which presented enhanced activity under high currents (up to 300 mA cm⁻²) while maintaining high H₂O₂ selectivity (~ 90%). However, there are still many problems to be clarified and some experiments are lack of. A substantial modifications are needed.

Response #3: We greatly appreciate the reviewer's valuable comments which have helped to greatly improve the quality of our study. We have now included additional experiments and analysis to fully address the reviewer's questions and concerns as follows:

1. Page 6 - “non-metal elements with atomic ratio of 0.41, 0.86, 0.34, and 0.90 at.% for boron, nitrogen, phosphorous and sulfur dopant, respectively”. So that the content of heteroatom in the catalysts was different. It is difficult to compare the performance since the catalyst loading is different.

2. Is it the characteristics of heteroatom or the content of heteroatom resulted in the difference of ORR activity? Please eliminate the influence of content of heteroatom content.

Response #3.1&3.2:

Thank you for your constructive comments here. We believe these two questions are related therefore we address them together in the following response.

We do agree that there is a difference in the doping level that we got for our series samples because we used a general and very facile method for the synthesis and the amount of the precursor is excessive. In fact, it is extremely difficult and unlikely to make all the dopants at exactly the same doping level in the carbon lattice. However, even without the same doping levels, it is still straightforward for us to make comparisons and convincing conclusions. As shown in Figure 1, the RRDE analysis exhibits that even some of the dopants have slightly higher ratio, e.g. nitrogen in N-C and sulphur in S-C, the performance of those two samples are even worse than B-C with a lower dopant loading. Thus, it is believed that the difference in dopant levels does not play a direct and important role here. Furthermore, we have slightly modified our synthetic procedures to introduce more dopants (B, N, P, and S) into carbon black while keeping the content of O at nearly the same level as previous samples. We have included the atomic ratio for each of the new samples in Table R2 and also in SI Table S7. In terms of the performance, the trend for all the doped samples are the same compared to the RRDE performance (Figure R4). Specifically, B-C-36 has the best kinetics and onset potential among all the four dopants, with N-C-36 having the 2nd best kinetics but lowest FE towards H₂O₂ production, followed by P-C-36 and S-C-36 with similar kinetics. A more convincing comparison can be made between our previous B-C (0.41% of B) and newly synthesized N-C-36 (1.12%), S-C-36 (1.06%) and P-C-36 (0.7%), where its B doping level is lower than all of other dopants but its 2e⁻-ORR performance is much higher than all other high doping level catalysts (Figure R4). Therefore, we can safely conclude that it is the type of dopant species,

not their concentrations, that play the most important role in determining the ORR to H_2O_2 performances. Figure R4 has also been included into revised SI (Figure S32).

Table R2. Atomic ratios of X-C-36 (X represents B, N, P and S) series catalysts determined by XPS.

Atomic Ratio	C (at. %)	Heteroatom (at. %)	O (at. %)
B-C-36	98.24	0.91	0.85
N-C-36	98.00	1.12	0.88
P-C-36	98.38	0.70	0.92
S-C-36	98.42	1.06	0.52

Figure R4. Three-electrode flow cell performance of catalysts. (a) I-V curve for B-C, B-C-36, N-C-36, P-C-36 and S-C-36 in 1 M Na_2SO_4 **(b)** Corresponding faradaic efficiencies measured. All the I-V curves are manually iR -compensated. B-C-36 shows highest selectivity among all the dopants, with the best activity as well. Furthermore, B-C with lower B loading still shows superior ORR performance compared to other dopants in the X-C-36 (X stands for N, P and S here) series with higher dopant level.

3. As shown in Fig. 1, the selectivity of H_2O_2 by P-C was higher than other four catalysts although the onset potentials was lower than B-C. However, the author's conclusion was B-C was best. Therefore, what is the decisive factor? And the how to choose this factor?

Response #3.3: Thank you for your invaluable comment here. We agree that in Figure 1, P-C shows similar selectivity for H_2O_2 compared B-C. However, that does not contradict with our conclusion in this work. The major challenge that this work is trying to resolve is to enhance

the catalytic activity to produce H_2O_2 under large currents while not sacrificing the selectivity. **The major factor to be considered is the catalyst's kinetics and overpotentials while maintaining a good H_2O_2 selectivity.** Thus, even though P-C gets similar selectivity to B-C in RRDE, its onset and activity is not as good as B-C, which could be the same case under large operation current densities. To further validate our hypothesis, we performed additional $2e^-$ -ORR evaluation of P-C in the flow cell reactor for a direct comparison with B-C. We can see that for large current regions, B-C shares slightly better selectivity than P-C (Figure R8), while exhibits much better kinetics and smaller overpotentials. We have also shown below the partial currents for H_2O_2 for both B-C and P-C. It is obvious that the performance for B-C at large currents are superior. **In conclusion, the decisive factor is the H_2O_2 activity under significant currents while maintaining high selectivity.**

Figure R8. B-C-36 and P-C-36 three-electrode flow cell performance in 1M Na₂SO₄. (a) faradaic efficiency and (b) H_2O_2 partial current. Note that the selectivity for B-C-36 is slightly better and the H_2O_2 partial current of B-C-36 is much higher compared to P-C-36.

4. Even if B-C was the best, have they investigated the effects of B doping amount?

Response #3.4: Thank you for your constructive comments. We agree that we should also study the effect of B doping amount. Thus, we have slightly modified our synthesis protocol and managed to get a higher B doping amount (see Table R5 below and also Table S6). Detailed synthetic method is included in our Methods section. Compared to the original B-C catalyst, the synthesized B-C-36 catalyst shows roughly doubled amount of boron. Specifically, B increases to 0.91 at% in B-C-36 from 0.41 at% in B-C. In terms of the performance, the kinetics for B-C with higher B doping amount shows significantly improved I-V curve while maintaining similar (or even slightly better) H_2O_2 selectivity. From Figure R9, we can see that

at 0.307 V vs. RHE, our original B-C reaches to an ORR current of 103 mA cm⁻² while the B-C-36 sample with higher B doping delivers a significantly improved current of 200 mA cm⁻². We want to note here that while it is extremely difficult and unlikely for us to maintain exactly the same level of oxygen species (which are mainly from the surface adsorbed water, please see our response #1.1 to Reviewer 1). Still, B-C-36 showed lower oxygen contents than B-C, which successfully excludes the possible positive impacts from the trace amount of oxygen species on its improved 2e⁻-ORR performances. Therefore, it is very clear that the increased concentration of B dopant is responsible for the enhanced ORR performance. The information in Figure R9 have been included into revised SI (Figure S31).

Table R5. Atomic ratios of B-C and B-C-36 determined by XPS.

Atomic Ratio	C (at. %)	B (at. %)	O (at. %)
B-C	98.34	0.41	1.25
B-C-36	98.24	0.91	0.85

Figure R9. Three-electrode flow cell performance of B-C with different loadings in 1M Na₂SO₄. (a) I-V curve and (b) Corresponding faradaic efficiencies measured. Note that all the I-V curves and faradaic efficiency were taken average of 2 independent tests for each of the samples. All the I-V curves are manually *iR*-compensated.

5. Some more details information on experiments are missing. For example, how the cathode was prepared using the B-C catalyst.

Response #3.5: Thank you for your invaluable comments. We have added more details in the method section (Page 26). The cathode electrode was prepared by uniformly spray coating the well-mixed catalyst ink on Sigracet 39 BC GDL electrode with the loading of B-C around 0.5 mg cm^{-2} . The ink is prepared by adding B-C powders and mixed with $4 \text{ }\mu\text{L mg}^{-1}$ binder (nafion 117), together with enough isopropanol as the solvent, and sonicate for 30 mins until uniformly dispersed.

6. In DFT calculations, the graphene was used as carbon structure in configurations of heteroatom doping carbon black, is this appropriate?

Response #3.6: Thank you very much for your important comment here. This is really a good question. Actually, the structure of carbon black (<https://doi.org/10.1016/B978-044451140-9/50013-5>) is comparable to graphite: both are composed of graphene sheets, while graphite's layers are typically larger and more ordered than carbon black whose sheets form 3-dimensional structure. Also, from the high resolution TEM image (Figure R10 shown below) for our carbon black catalyst, the regional structure for carbon black is graphene-like structure. Based on this information, graphene structure is typically used to represent carbon black for modelling electrochemistry. (<https://pubs.rsc.org/en/content/articlehtml/2019/sc/c8sc05236k>; <https://www.nature.com/articles/s41524-019-0210-3.pdf?origin=ppub>). Considering the 3-dimensional structure of carbon black would be too large for DFT modeling while chemical reaction typically takes place at $\sim 1 \text{ nm}$ scale, we believe that using graphene as the carbon structure in our simulations is appropriate to our best computational resources. Figure R10 have been included into revised SI (Figure S17).

Figure R10. High resolution TEM image of CB. Note that the local structure in nano-meter scale is exactly graphene-like structure.

7. In Fig. 5b and 5c, the Higher H_2O_2 activities by B-C was showed compared with O-C, however, the FE was lower. Why?

Response #3.7: Thank you for your invaluable comment. Typically, activity and FE (selectivity) do not always change in the same direction. This goes back to the problems associated with the currently reported oxidized carbon materials: they have high FEs, but their kinetics under high currents are not good so their overpotentials are large. The reason we report B-C in this work is because it can greatly reduce the overpotential that O-C needs to deliver high currents (improve the kinetics), while still maintaining similarly high FEs. Thus, in terms of the large-scale long-term H_2O_2 production, it is much worthy to achieve a higher activity while not sacrificing much of FEs to save a significant amount of energy. Moreover, if look at the H_2O_2 partial current densities, B-C is much superior to that of O-C (Figure 5b and 5e in the manuscript). Therefore, although the FE for B-C is slightly lower than that of O-C, B-C is still a superior catalyst to produce H_2O_2 in a large scale due to its better activity under high H_2O_2 production rate.

8. Page 17 - “With the incorporation of our B-C catalyst into this cell configuration as the anode, under a fixed DI water feed rate” Should the B-C catalyst be used as cathode?

Response #3.8: Thank you for your invaluable comments. Yes, it is a typo here. B-C should be the cathode, as the working electrode for ORR. Sorry for the confusion and we appreciate that you pointed this out. The corresponding wordings have been fixed in both the manuscript and the supplementary information.

9. For prepared B-C, why did the remnant need to be annealed, however, other catalysts are not needed?

Response #3.9: Thank you for your invaluable comments. For B-C, since we use boric acid as the precursor, there is an extra step of using hot water to remove the B_2O_3 from the annealing remnant because B_2O_3 is not volatile (<https://doi.org/10.1016/j.jallcom.2010.02.114>). The extra annealing process is to make sure that the samples are not further oxidized during this process to remove later-introduced oxygen species. For other dopants, since the precursor does not generate non-volatile impurities, there is no need for such an extra step.

10. In Fig. 5g, it is suggested to provide generation of H_2O_2 at different pH conditions.

Response #3.10: Thank you for your invaluable comments. We agree that for different application scenarios in industry, H_2O_2 with different pHs should be considered. 30 hours of stability test have already been demonstrated in 1 M KOH at different currents mimicking different application scenarios (200 mA cm^{-2} in Figure 5g and 60mA cm^{-2} in Figure S29). Furthermore, we have performed the stability test in 1 M Na_2SO_4 solution for 30 hours (see Figure R11 and Figure S30). Furthermore, we have performed a 200-hour stable production of 1100 ppm pure H_2O_2 solutions, as shown in Figure R12 and Figure 6d (and Figure S33 for zoom-in view for first 10 hours), to further demonstrate the superior potential of the catalyst to be used in large-scale industrial practical applications.

Figure R11. Stability performance test for B-C in 1 M Na₂SO₄ in three-electrode flow cell configuration. The current is fixed at 60 mA cm⁻², with the electrolyte feeding rate fixed at 54 mL h⁻¹ and oxygen feeding rate fixed at 20 sccm. The catalyst retains its catalytic activity and faradaic efficiency for 30 hours without any degradation, indicating its excellent stability.

Figure R12. Long-term pure H₂O₂ production of B-C in solid-electrolyte cell configuration. (a) Stability test of B-C fixed at 30 mA cm⁻² of generation of ~1,100 ppm pure H₂O₂ solution. **(b)** Zoom-in view of first 10 hours. The DI water feeding rate is fixed at 54 mL h⁻¹. The catalyst retains its catalytic activity and faradaic efficiency for 200 hours without any degradation, indicating its excellent stability and great potential in future large-scale practical applications.

11. In Table S4, it would be more convinced to compare more non-metal doped carbon materials.

Response #3.11: Thank you for your invaluable comments. We have added more non-metal doped carbon materials in Table S4. Even compared to recently reported hetero-atom doping materials, our B-C still shows the best performance. This table have been included into revised SI (Table S4).

Table R3. Comparison of electrochemical O₂-to-H₂O₂ performance under high currents with state-of-the-art catalysts.

Catalyst	Electrolyte	Potential	j_{total} (mA cm ⁻²)	FE (%)	Stability
This work	1M KOH	0.685 V vs. RHE	300	85.1	30 hrs @ 200 mA cm⁻²
	1M Na₂SO₄	0.277 V vs. RHE	300	83.2	30 hrs @ 60 mA cm⁻²
	DI water (Solid-electrolyte reactor)	2.55 V	400	85.5	200 hrs @ 30 mA cm⁻²
Fe-O-CNT ³	1M KOH	0.76 V vs. RHE	45	95.4	8 hrs @ 1-3 mA cm ⁻²
Natural air diffusion electrode ⁴	0.05 M Na ₂ SO ₄	Not mentioned	240	66.8	20 hrs @ 60mA cm ⁻²
O-CNT ⁵	1M KOH	0.68 V vs. RHE	40	90	10 hrs @ 0.2-0.4 mA
Ni-N ₂ O ₂ /C ⁶	0.1M KOH	0.5 V vs. RHE (without iR -compensation)	70	91	8 hrs @ 70 mA cm ⁻²
Co-N-C ⁷	0.1M KOH	~0.55 V vs. RHE	50	Not mentioned	110 hrs @ ~2.4 mA cm ⁻²
C-PTFE electrode ⁸	0.05M Na ₂ SO ₄	Not mentioned	145	29	N/A
PtP ₂ -Al ₂ O ₃ ⁹	membrane fuel cell (H ₂ anode)	Not mentioned	150	78.8	110 hrs @ 0.4V (current density not mention)

N-doped mesoporous carbon ¹⁰	0.1M KOH	0.3 V vs. RHE	2 in RRDE	82	6 hrs @ 3 mA cm ⁻²
	0.1M K ₂ SO ₄	0.2 V vs. RHE	3 in RRDE	75	6 hrs @ 3.5 mA cm ⁻²
B,N doped carbon ¹¹	0.1M KOH	0.55 V vs. RHE	1.5 in RRDE	85	50 hrs @ ~1.2 mA cm ⁻²
N-doped carbon nanohorns ¹²	0.1M NaOH	0.65 V vs. RHE	~0.6 in RRDE	~65	25 hrs @ ~0.6 mA cm ⁻²
	0.1M PBS	0.45 V vs. RHE	~0.5 in RRDE	~90	25 hrs @ ~0.5 mA cm ⁻²

- 3 Jiang, K. *et al.* Highly selective oxygen reduction to hydrogen peroxide on transition metal single atom coordination. *Nature communications* **10**, 1-11 (2019).
- 4 Zhang, Q. *et al.* Highly efficient electrosynthesis of hydrogen peroxide on a superhydrophobic three-phase interface by natural air diffusion. *Nature communications* **11**, 1-11 (2020).
- 5 Lu, Z. *et al.* High-efficiency oxygen reduction to hydrogen peroxide catalysed by oxidized carbon materials. *Nature Catalysis* **1**, 156-162 (2018).
- 6 Wang, Y. *et al.* High-Efficiency Oxygen Reduction to Hydrogen Peroxide Catalyzed by Nickel Single-Atom Catalysts with Tetradentate N₂O₂ Coordination in a Three-Phase Flow Cell. *Angewandte Chemie International Edition* (2020).
- 7 Jung, E. *et al.* Atomic-level tuning of Co–N–C catalyst for high-performance electrochemical H₂O₂ production. *Nature Materials* **19**, 436-442 (2020).
- 8 Brillas, E., Calpe, J. C. & Casado, J. Mineralization of 2, 4-D by advanced electrochemical oxidation processes. *Water Research* **34**, 2253-2262 (2000).
- 9 Li, H. *et al.* Scalable neutral H₂O₂ electrosynthesis by platinum diphosphide nanocrystals by regulating oxygen reduction reaction pathways. *Nature communications* **11**, 1-12 (2020).
- 10 Sun, Y. *et al.* Efficient Electrochemical Hydrogen Peroxide Production from Molecular Oxygen on Nitrogen-Doped Mesoporous Carbon Catalysts. *ACS Catalysis* **8**, 2844-2856, doi:10.1021/acscatal.7b03464 (2018).
- 11 Chen, S. *et al.* Designing boron nitride islands in carbon materials for efficient electrochemical synthesis of hydrogen peroxide. *Journal of the American Chemical Society* **140**, 7851-7859 (2018).
- 12 Iglesias, D. *et al.* N-doped graphitized carbon nanohorns as a forefront electrocatalyst in highly selective O₂ reduction to H₂O₂. *Chem* **4**, 106-123 (2018).

12. For industrial application, 25 h stability test is not sufficient.

Response #3.12: Thank you for your invaluable comments. We agree that 25 hours are not enough for large-scale industrial applications. Thus, we did another stability test for 200 hours of pure H₂O₂ solution production and the performance shows no degradation in activity and selectivity, indicating the promising stability of our B-C catalyst.

Figure R12. Long-term pure H₂O₂ production of B-C in solid-electrolyte cell configuration. (a) Stability test of B-C fixed at 30 mA cm⁻² of generation of ~1,100 ppm pure H₂O₂ solution. **(b)** Zoom-in view of first 10 hours. The DI water feeding rate is fixed at 54 mL h⁻¹. The catalyst retains its catalytic activity and faradaic efficiency for 200 hours without any degradation, indicating its excellent stability and great potential in future large-scale practical applications.

13. In solid electrolyte cell, what effect if not using 1 M H₂SO₄?

Response #3.13: Thank you for your invaluable comments. 1 M H₂SO₄ was used for IrO₂, because it was reported that IrO₂ is more stable in acidic OER conditions (<https://doi.org/10.1073/pnas.1915319116>). However, after we replaced anolyte of H₂SO₄ by pure DI water, the whole reaction is still stable and can be operated for over 200 hours. Thus, DI water can also be used as the anolyte in our solid electrolyte design. The 200-hour performance is also shown above as Figure R12.

14. It is suggest to give energy consumption analysis or H₂O₂ generation cost to check the advantage of this method.

Response #3.14: Thank you for your invaluable comments. We agree that it is more informative if we provide a direct comparison in terms of the energy cost. Thus, we have provided it in Supplementary note 1 and also attach it as follows:

In the following paragraph, we have performed a simple analysis for energy cost. Please note that the only cost we consider at this stage are energy and feedstock cost (oxygen, electricity, DI water, etc.) but without any equipment cost, i.e. operation cost.

If we are operating the solid-electrolyte cell under highest H₂O₂ production rate condition (see **Figure 6c**): 2.55 V (500 mA, ~0.84 wt.% H₂O₂), with a H₂O₂ production rate of 7.36 mmol cm⁻² h⁻¹ (1.00096 g h⁻¹). The mass of H₂O₂ generated using 1 kWh of electricity will be:

$$m_{H_2O_2} = \frac{1 \text{ kWh}}{2.55 \text{ V} * 0.5 \text{ A}} * 1.00096 \text{ g h}^{-1} = \sim 785.1 \text{ g}$$

- (a) The as-generated 0.7851 kg H₂O₂ consumes 0.7389 kg O₂ from stoichiometric balance. If the industrial O₂ price is set to be about < \$0.1/kg (source: <https://www.intratec.us/chemical-markets/oxygen-price>), the total cost of O₂ feed can be summed as 7.39 cents (**assuming \$0.1/kg-H₂O₂ for O₂ price**; but in reality, the cost is even lower). Please note that the O₂ cost can be further reduced by collecting and recycling O₂ gas produced from OER on the anode side.
- (b) Assuming the price of electricity is 3 cents/kWh (*Nature Materials* **16**, 16–22 (2017)), we can roughly estimate a electricity for H₂O₂ production to be **\$0.038/kg-H₂O₂**.
- (c) Since we locate in Texas, US, we just use the water price in Texas as our calculation basis. The industrial water in Texas is about \$1.91/(1000 gallon), or \$0.00191/gallon (<https://www.fbgtx.org/673/Industrial-Water-Rates>). The price for deionize one gallon of water is lower than \$0.03 (<https://blog.uswatersystems.com/2012/08/de-ionization-101/>). Thus, the total cost for DI water is less than about \$0.03/gallon, i.e. **\$0.008/kg**, or about **\$0.9/kg-H₂O₂**.

In summary, the total energy and feed stock cost is less than about $\$1/\text{kg-H}_2\text{O}_2$. Since the largest portion of the cost is from deionizing water, the cost can be further reduced in the future by replacing DI water feed stock with industrial water plus water filter. Furthermore, our on-site generation method does not need the cost for transportation and storage. In comparison, the traditional industrial anthraquinone process for H_2O_2 production has a rough cost of $\$1.5/\text{kg-H}_2\text{O}_2$ without transportation and storage cost (<http://www.h2o2.com/faqs/FaqDetail.aspx?fid=25>). Thus, our method is still far more economical than the traditional way.

Reviewer #3 (Remarks to the Author):

The revision of the paper has addressed most of the questions, however, the following items still need improvement before acceptance.

- 1) The H₂O₂ generation cost. The calculation method should be modified because the following: 1) actually, oxygen can not be completely used for hydrogen peroxide electrosynthesis, i.e., O₂ calculation should not be from stoichiometric balance. 2) The energy consumption on sparging oxygen should also be taken into consideration.
- 2) For actual H₂O₂ production, the electrolyte flow rate was only 54 mL/h, can it be scale up in viewpoint of application?

Title: “**Highly Active and Selective Oxygen Reduction to H₂O₂ on Boron-Doped Carbon for High Production Rates**”

Authors: Yang Xia, Xunhua Zhao, Chuan Xia, Zhen-Yu Wu, Peng Zhu, Jung Yoon (Timothy) Kim, Xiaowan Bai, Guanhui Gao, Yongfeng Hu, Jun Zhong, Yuanyue Liu, Haotian Wang

Corresponding authors: Yuanyue Liu, Haotian Wang

We thank the reviewers for the constructive comments which have helped us to greatly improve our research and the quality of our manuscript. We have now included additional analysis and discussions to fully address the reviewers’ concerns and suggestions. **Furthermore, we have modified our manuscript and SI based on the additional results and analysis, which are highlighted in yellow.** Below, we address the points raised by reviewers one by one.

Reviewer 3

The revision of the paper has addressed most of the questions, however, the following items still need improvement before acceptance.

Response # 3: We greatly appreciate the reviewer’s valuable comments which have helped to greatly improve the quality of our study. We have now included additional analysis to fully address the reviewer’s questions and concerns as follows:

(1) The H₂O₂ generation cost. The calculation method should be modified because the following: 1) actually, oxygen can not be completely used for hydrogen peroxide electrosynthesis, i.e., O₂ calculation should not be from stoichiometric balance.

Response # 3.1: Thank you for your important comments here. We completely agree that the calculation of oxygen cost is not accurate enough if that is based on the stoichiometric balance, since the reactor is typically supplied by some excess of oxygen. Thus, we have modified the calculation and added the following part to our supplementary note together with the previous calculations:

In our actual test, the flow rate of O₂ feed is 20 sccm (20 standard cubic centimeters per minute), which corresponds to 1200 cm³ hr⁻¹, or 1.2 L hr⁻¹. **We can use one hour as the time basis for a simple calculation.** Based on the Ideal Gas Law, i.e. PV=nRT, at standard condition (assuming T=273.15K, P=1atm, R=0.0821(L*atm/(mol*K))) with V=1.2L, n=0.0535mol, which gives the mass of O₂ supply to be:

$$\text{mass of } O_2 = 0.0535 \text{ mol of } O_2 \times \frac{32 \text{ g}}{\text{mol}} = 1.71 \text{ g } O_2$$

From our previous calculation, at 2.55 V (500 mA, ~0.84 wt.% H₂O₂), the H₂O₂ production rate for one hour is 7.36 mmol cm⁻², i.e. 1.00096 g. Thus, the O₂ cost per mass of H₂O₂ produced is:

$$\text{The mass cost of } O_2 = \frac{1.71 \text{ g } O_2}{1.00096 \text{ g } H_2O_2} = \frac{1.708 \text{ g } O_2}{\text{g } H_2O_2 \text{ produced}} = 1.708 \text{ kg } O_2/\text{kg } H_2O_2 \text{ produced}$$

Given the industrial O₂ price set to be about < \$0.1/kg (source: <https://www.intratec.us/chemical-markets/oxygen-price>), the total cost of O₂ feed can be summed as 0.1708 dollars (17.1 cents) (**assuming \$0.17/kg-H₂O₂ for O₂ price**; but in reality, the cost is even lower). Please note that the O₂ cost can be further reduced by collecting and recycling O₂ gas produced from OER on the anode side.

Note that even adding the cost of the excessive O₂ supply, it only adds a marginal cost of 7 cents per kilograms of H₂O₂ generated (**\$0.17/kg-H₂O₂** based on actual excessive usage of O₂) compared to the previous calculation we performed in our latest response letter if calculating the usage of O₂ based on stoichiometric balance only (**\$0.1/kg-H₂O₂**), which counts a small portion of the total cost (**\$1.1/kg-H₂O₂** adding the cost of O₂, electricity and DI water together).

2) The energy consumption on sparging oxygen should also be taken into consideration.

Response # 3.2: Thank you for your important comments here. We agree that in real industrial application, there is a step for sparging oxygen, which requires some cost for the mechanical energy. However, in our real experiments, the oxygen is supplied already with high pressure (in commercial gas cylinders). Thus, the sparging cost is already calculated as a part of the O₂ price and there is no additional need for a separate sparging oxygen cost calculation.

(2) For actual H₂O₂ production, the electrolyte flow rate was only 54 mL/h, can it be scale up in viewpoint of application?

Response # 3.3: Thank you for your important comments here. The actual electrolyte flow rate of 54 mL/h that we use in our experiment corresponds to the 4 cm² electrode, which gives a surface flowrate of 13.5 mL h⁻¹ cm⁻². In the viewpoint of scale-up application, for example a device with 100-1000 cm² electrode, the output flowrate can be scaled up to 1.35 to 13.5 L/h per unit cell and can be further scaled up when stacking multiple cells together.